# Probing phonon transport dynamics across an interface by electron microscopy

Fachen Liu[1,2,3,8], Ruilin Mao[1,2,8], Zhiqiang Liu[4], Jinlong Du[2] & Peng Gao[1,2,3,5,6,7 ✉]

Understanding thermal transport mechanisms across material interfaces is crucial for advancing semiconductor technologies, particularly in miniaturized devices operating under extreme power densities[1,2]. Although the interface phonon-mediated processes are theoretically established[3–6] as the dominant mechanism for interfacial thermal transport in semiconductors[7], their nanoscale dynamics remain experimentally elusive owing to challenges in measuring the temperature and non-equilibrium phonon distributions across the buried interface[8–11]. Here we overcome these limitations by using in situ vibrational electron energy-loss spectroscopy (EELS) in an electron microscope to nanoscale profile temperature gradients across the AlN–SiC interface during thermal transport and map its non-equilibrium phonon occupations at sub-nanometre resolution. We observe a sharp temperature drop within about 2 nm across the interface, enabling direct extraction of relative interfacial thermal resistance (ITR). During thermal transport, the mismatch of phonon modes' thermal conductivity at the interface causes substantial non-equilibrium phonons nearby, making the populations of interface modes different under forward and reverse heat flow and also leading to marked changes in the modal temperature of AlN optical phonons within about 3 nm of the interface. These results reveal the phonon transport dynamics at the (sub-)nanoscale and establish the inelastic phonon scattering mechanism involved by interface modes, offering valuable insights into the engineering of thermal interfaces.

The rapid development of information technologies, such as artificial intelligence and big data, has increased demands for semiconductor thermal management[1]. As devices shrink and power increases, semiconductor interfaces dominate thermal resistance[1,2], causing elevated interface temperatures that degrade the device performance and reliability[1,12] and thus posing notable challenges in on-chip thermal management[13]. In these semiconductor materials, phonons are the primary heat carriers. At the interface, mismatches in phonon energy and momentum owing to discontinuities in chemical bonding[14], elemental composition[15] and symmetry lead to substantial phonon scattering, thus increasing thermal resistance and intensifying the hotspots. Especially in the transistor drain region, nanoscale hotspots[16] originate from the localized accumulation and far-from-equilibrium behaviour of slow optical phonons[2] owing to exacerbated phonon scattering from the interface[2,17]. Therefore, studying the non-equilibrium phonon transport at interfaces is necessary.

Phonon transport in solids is typically driven by temperature gradient. The ratio of the temperature drop across the interface to the heat flux is known as the ITR and is used to characterize the thermal transport properties of an interface, which has received a lot of attention in the past century[18–20]. As well as computational tools such as non-equilibrium molecular dynamics (NEMD)[5,6,21] and Boltzmann transport equations[19,22],

the study of ITR encompasses experimental methods such as time-domain thermal reflectance[23,24] and frequency-domain thermal reflectance[25]. These macroscopic experimental methods measure thermal diffusion coefficients or thermal resistance but they average out contributions from interface roughness, interfacial intermediate layers and other structural inhomogeneity far away from the interface, such as nanoscale/atomic-scale defects in the bulk owing to their lack of spatial resolution[26] and structural characterization capability. At the nanoscale, surface probe techniques such as scanning thermal microscopy[27] and tip-enhanced Raman spectroscopy[28] can obtain the surface temperature and electron self-heating in scanning electron microscopy achieves thermal resistance mapping at approximately 20 nm resolution[29]. Nevertheless, the phonon dynamics across buried interfaces during thermal transport still remain poorly understood, leaving several important predictions unverified, such as the intrinsic interface temperature drop width[3,4], the temperature-gradient-induced non-equilibrium phonon distribution at the interface[5,6] and evolution of interface phonons during heat transfer[30].

Recent advances in scanning transmission electron microscopy electron energy-loss spectroscopy (STEM-EELS) enable direct detection of equilibrium localized heterointerface phonons and defect phonons[11,15,31–36] at room temperature, which are sensitive to the chemical

[1]International Center for Quantum Materials, School of Physics, Peking University, Beijing, China. [2]Electron Microscopy Laboratory, School of Physics, Peking University, Beijing, China. [3]Academy for Advanced Interdisciplinary Studies, Peking University, Beijing, China. [4]Research and Development Center for Solid State Lighting, Institute of Semiconductors, Chinese Academy of Sciences, Beijing, China. [5]Tsientang Institute for Advanced Study, Hangzhou, China. [6]Interdisciplinary Institute of Light-Element Quantum Materials and Research Center for Light-Element Advanced Materials, Peking University, Beijing, China. [7]Collaborative Innovation Center of Quantum Matter, Beijing, China. [8]These authors contributed equally: Fachen Liu, Ruilin Mao. ✉e-mail: pgao@pku.edu.cn

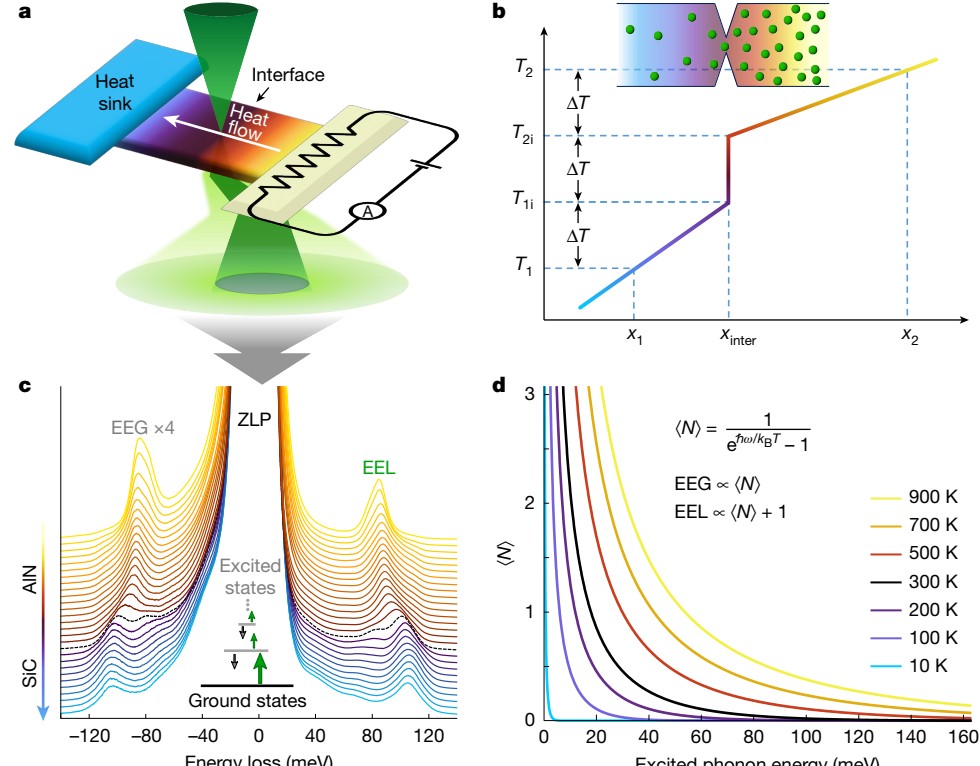

**Fig. 1 | In situ STEM-EELS for probing phonon transport dynamics across an interface. a**, Schematic of the experimental set-up for in situ heating STEM-EELS, depicting an AlN–SiC heterostructure with a heated (hot) side and a cooled (cold) side to establish a temperature gradient across the interface. **b**, Schematic of the temperature distribution near the interface in an idealized one-dimensional steady-state heat conduction model, in which temperature drop ($\Delta T = T_{2i} - T_{1i}$) at the interface is proportional to the ITR. The characteristic interface length[20,44], which is defined as the spatial distance over which the bulk thermal resistance equals the ITR, is shown as ($x_2 - x_{inter}$) on the right side and ($x_{inter} - x_1$) on the left.

**c**, EEL and EEG spectra acquired from hot AlN to cold SiC. The black dashed spectrum was acquired at the interface. For better visualization, the spectra were normalized to EEL signal intensity, with EEG spectra scaled by a factor of four. The inset shows that the EEL signal mainly represents the phonon ground state, whereas the EEG signal reflects thermally excited state phonons. **d**, Average excited state phonon population $\langle N \rangle$ as the function of temperature and energy. The EEG signal intensity directly quantifies $\langle N \rangle$ (thermally excited state phonons), whereas the EEL signal intensity corresponds to the total phonon population $1 + \langle N \rangle$ (ground state + thermally excited state).

bond[37,38], interface roughness[11], composition/isotope variation[31,39] and local atomic configurations[15], providing important insights into ITR. However, investigating non-equilibrium phonon transport behaviour across interfaces necessitates establishing large temperature gradients at the interface during STEM-EELS measurements. Moreover, it requires the information of phonon populations, which deviate from the Bose–Einstein distribution in non-equilibrium states. Phonon populations can be directly reflected in electron energy loss/gain (EEL/EEG) signals by the principle of detailed balancing and can be used for nanoscale temperature measurements[8–10]. At interfaces with temperature gradient, notable non-equilibrium phonon populations are expected owing to changes of transmission coefficients for different phonon modes (that is, the interface acts as a filter for redistributing these populations[6]). Therefore, analysing local EEL/EEG signals near the interface during thermal transport can extract the non-equilibrium phonon populations, explaining the phonon transport behaviour at the interface.

In this work, we study the phonon transport dynamics at the AlN–SiC interface with sub-nanometre resolution by using in situ vibrational STEM-EELS. A stable temperature gradient of about 180 K μm⁻¹ is generated at a thin foil heterointerface for STEM-EELS characterization. From the (sub-)nanometre-resolution temperature map, the temperature drop at the approximately 2-nm-wide interface, with a magnitude of about 10–20 K, is extracted and compared with that in the bulk to extract the relative ITR, whose characteristic length spans from tens to hundreds of nanometres at different temperatures. The EEG signals show that the interface scattering leads to substantial non-equilibrium phonons within roughly 3 nm near the interface, further altering the

nearby AlN optical phonon modal temperature. Heat flow direction affects the non-equilibrium phonon populations of interface modes: the α/β mode at 75/90 meV is substantially populated under forward/reverse heat flow, respectively. Combined with mode-resolved NEMD simulations, we suggest that, during inelastic scattering, interface phonons preferentially couple with higher-energy bulk phonons, regardless of hotter or colder side. Our work demonstrates that this approach, phonon-transport electron microscopy, is powerful for studying nanoscale thermal transport, especially for buried interfaces and structural defects. The findings of non-equilibrium phonon populations and transport dynamics across an interface provide valuable insights into interfacial thermal transport mechanisms and useful information for the thermal management of semiconductor devices.

Figure 1a shows a locally heated AlN–SiC heterostructure in a STEM sample, generating a notable temperature gradient at the interface (Fig. 1b) (see Methods and Supplementary Fig. 1 for details). During in situ EELS measurements, EEL and EEG signals are acquired simultaneously (Fig. 1c). The EEG signal intensity directly reflects the population of thermally excited states $\langle N \rangle$ (Fig. 1d), whereas the EEL signal intensity represents the total number of excited and ground state phonons $1 + \langle N \rangle$, indicating that the EEG signal is more sensitive to the temperature. By quantitatively comparing the intensity ratio of EEG to EEL, we can map local temperatures[8–10] and thus the temperature drop at the interface, allowing us to estimate the ITR and correlate it with phonon transport behaviour. When the non-equilibrium system relaxed to steady state, the local temperature can be defined as time-average at nanoscale that is smaller than the mean free path (MFP) of phonons.

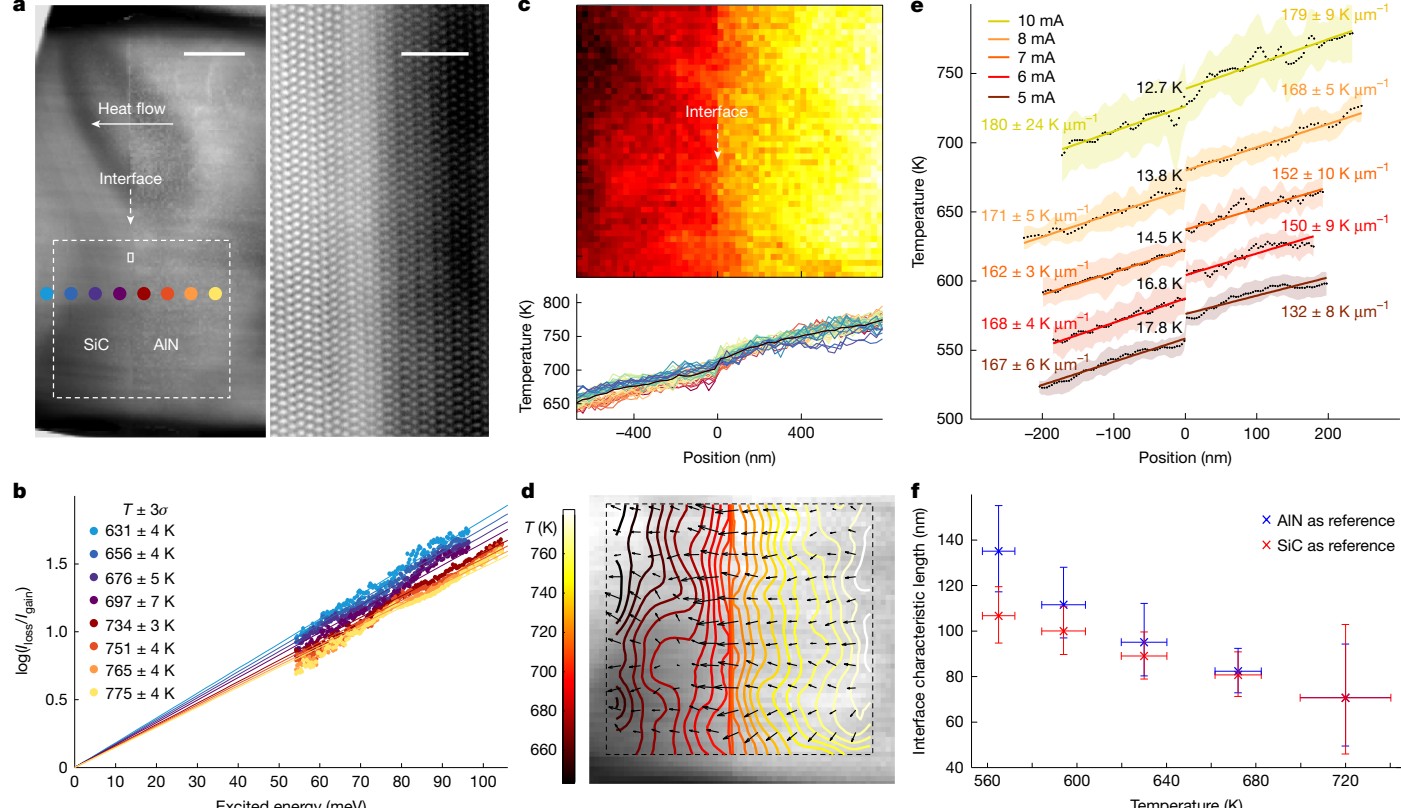

**Fig. 2 | Temperature map and ITR characterization under AlN-to-SiC heat flow. a**, Low-magnification high-angle annular dark-field STEM image (left) and atomic-resolution image (right) of the AlN–SiC interface. Scale bars, 500 nm (left), 2 nm (right). **b**, Linear plots of the logarithm of the ratio between loss and gain scattering as a function of excitation energy. Colour-coded spectra correspond to the acquisition positions marked by the corresponding coloured dots in **a**. **c**, Temperature map of the area marked by the white dashed box in **a** and the temperature profiles of each row. **d**, Corresponding isotherm diagram and temperature gradient field (black arrows), superimposed on the high-angle annular dark-field image to visualize thermal transport directionality. **e**, Interface-adjacent temperature profiles under varying heating currents from another sample (see Supplementary Fig. 1a). Linear fits to bulk regions (coloured text) yield temperature gradients, whereas the interface temperature drop ($\Delta T$) is denoted by black text. Coloured shaded areas represent the standard deviation of the mean from several data points. **f**, Relative ITR is quantified by the interface characteristic length, derived from the relationship between the interface temperature drop and the bulk temperature gradient in AlN and SiC, respectively.

To achieve nanoscale acquisition near the interface, we use the off-axis configuration[40] to enhance the localized non-dipolar EEL/EEG signal, thus allowing for spatially resolved temperature maps.

Without temperature gradient (and thus no net heat flow), bulk phonons and interface phonons of EELS remain nearly unchanged, except for a slight energy shift between AlN and SiC (Supplementary Fig. 2). Under a stable temperature gradient, temperature and heat flow can be mapped across the micron scale to the nanoscale. Figure 2a shows low-magnification and atomic-resolution images of the 4H-SiC/2H-AlN interface. Heat flow is designed to pass through the interface to conform to a one-dimensional heat transfer model. Figure 2b shows a linear fit with zero intercept for $\log(I_{\text{loss}}/I_{\text{gain}})$ at the eight selected positions marked in Fig. 2a. Under equilibrium conditions, we have:

$$\log\left(\frac{I_{\text{loss}}}{I_{\text{gain}}}\right) = \log\left(\frac{1 + \langle N \rangle}{\langle N \rangle}\right) \quad (1)$$

$$= \frac{1}{k_{\text{B}}T}\hbar\omega \quad (2)$$

in which $k_{\text{B}}$ is the Boltzmann factor and $\omega$ is phonon frequency. Theoretically, the fast electron–phonon interaction time is much smaller than the relaxation time of the phonon[41], ensuring that equation (1) always holds, which is also required by the principle of detailed balancing[9].

Under a temperature gradient, the phonon population $n$ differs slightly from $\langle N \rangle$; however, this minor deviation in our experiments has a negligible effect on the temperature calculations, meaning that equation (2) remains valid (see Supplementary Information Section 2 for details). In fact, the scatter points in Fig. 2b align closely with the fitted line, indicating that the deviation from equilibrium state is minimal at the micron scale.

From the selected region (dashed box in Fig. 2a), we mapped temperature distributions and plotted line profiles in Fig. 2c, revealing an approximately 140 K temperature drop over about 1.5 µm from the heated AlN to cold SiC. The map shows consistent temperature values across all lines, confirming that our method can produce a stable temperature gradient with an abrupt temperature change at the interface. The heat flow direction is determined by calculating the temperature field gradient (Fig. 2d), in which the isotherm distribution exhibits minimal parallel heat flow component to the interface, confirming quasi-one-dimensional heat transfer.

Figure 2e shows that, as the current increases, the overall temperature near the interface rises and the temperature gradient inside the bulk region also increases proportionally with the heat flow. Notably, the interface temperature drop decreases from 17.8 K to 12.7 K with rising temperature, indicating the reduction of ITR—a trend consistent with theoretical predictions from previous models[42,43]. Using the interface characteristic length[20,44] (defined in Fig. 1b), the relative ITR can be quantified by comparing the interface temperature drop with the bulk

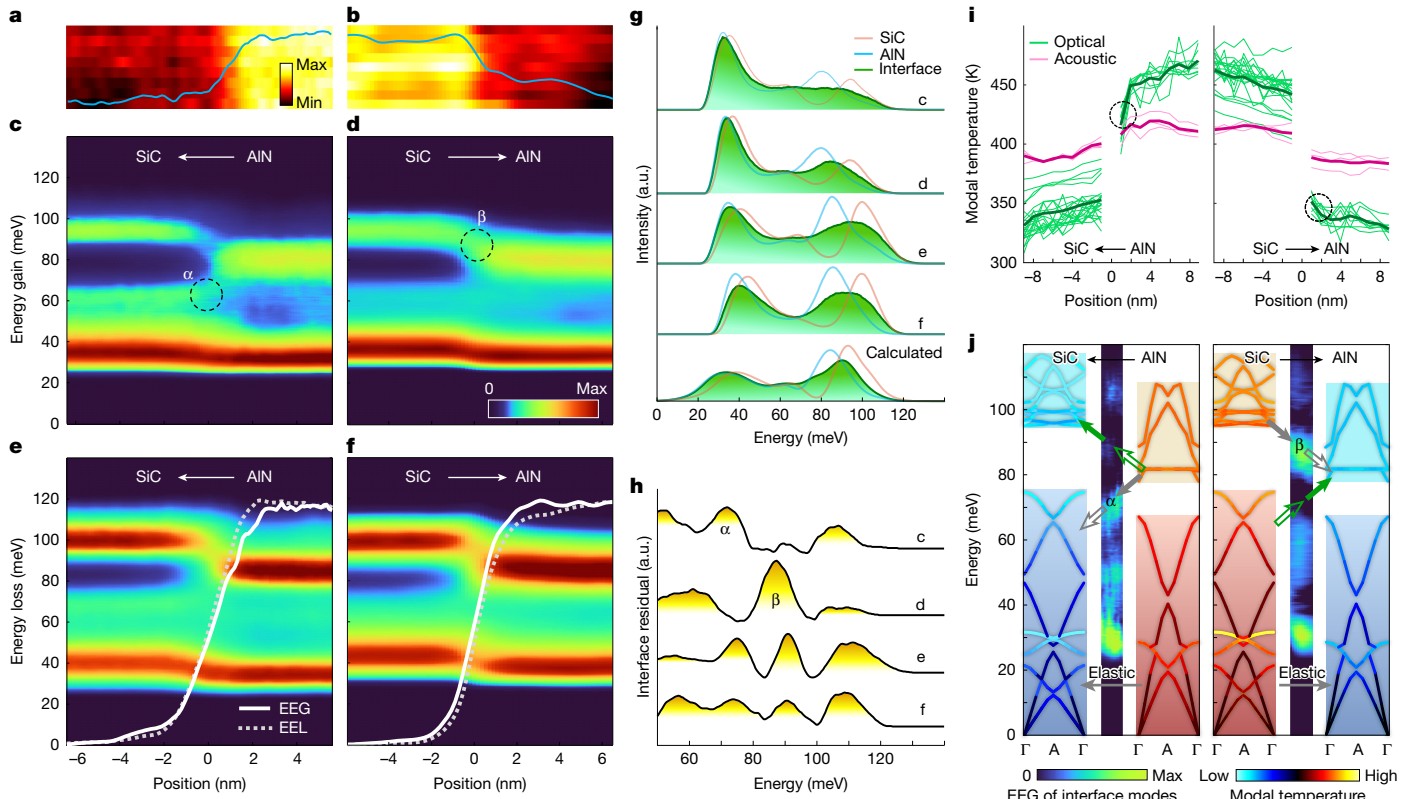

**Fig. 3 | Non-equilibrium phonon dynamics across the interface under forward (AlN → SiC) and reverse (SiC → AlN) heat flow. a,b,** Experimentally reconstructed temperature maps for forward and reverse heat flow conditions. **c,d,** EEG spectra acquired under forward and reverse heat flow. **e,f,** Corresponding EEL spectra. Overlaid lines denote the relative intensity of AlN TO phonon peak (81–85 meV): solid white line from EEG spectra and dashed grey line from EEL spectra. **g,** Extracted phonon spectra at SiC (red line), AlN (blue line) and interface (green shade) from panels **c**–**f**. Calculated phonon DOS is shown at the bottom. **h,** Interface residual spectra after removing the spectral components of bulk SiC and AlN using the least-squares fitting (see Methods). **i,** Calculated spatial distribution of phonon modal temperatures near the interface. The thick solid line represents the averaged modal temperature profile. **j,** Schematic illustration of non-equilibrium phonon transport across the interface. Dispersion lines are colour-coded by calculated modal temperatures along the ΓA direction (heat flow direction). The central coloured band plots depict the interface residual spectra from the experimental EEG data, representing the interface mode population. Arrows denote the three-phonon scattering processes associated with the α and β modes: green for absorption, grey for emission. Solid arrows indicate enhanced processes and open arrows denote suppressed processes. a.u., arbitrary units.

temperature gradient. Figure 2f shows that, as temperature increases from 565 to 720 K, the interface characteristic length decreases monotonically from about 107 nm to about 71 nm of SiC and from about 135 nm to about 71 nm of AlN.

Next, we investigate the dynamics of temperature-gradient-driven phonon transport across the interface by establishing forward (from AlN to SiC) and reverse (SiC to AlN) heat flow configurations. The temperature maps in Fig. 3a (forward) and Fig. 3b (reverse), derived from the optical phonons, show that temperature changes occur primarily within roughly 2 nm of the interface—consistent with the theoretical model[32]. Considering that the electron beam size of about 0.3 nm is much smaller than the length scale of temperature change, this measurement lies below the spatial resolution limit of the temperature map—a limit governed by the spatial characteristic length of phonon spectrum variation.

The broken symmetry at the interface generates unique interface phonon modes, as confirmed by previous EELS experiments[11,15,31–36]. Such interface modes are theoretically predicted to greatly influence thermal transport[6,30,45,46]. We analyse the evolution of interface phonon modes under both forward (AlN → SiC) and reverse (SiC → AlN) heat transfer, with corresponding EEG and EEL signals shown in Fig. 3c–f. Bulk and interface spectra, alongside calculated phonon density of states (DOS), are shown in Fig. 3g. EEL signals exhibit no notable difference between forward/reverse directions for either bulk or interface spectra, aligning closely with the calculated phonon DOS predictions.

By contrast, EEG signals show substantial differences in interface spectral features. For forward heat flow (Fig. 3c), the interface α mode (about 75 meV) exhibits much higher intensity than the interface β mode (about 90 meV); the reverse trend is observed for backward heat flow (Fig. 3d). The vibration eigenvectors of the interface modes α and β are shown in Supplementary Fig. 5. This phenomenon suggests a non-equilibrium population of interface modes during thermal transport. On comparing the finely extracted interface residual spectra that reflect pure localized interface modes in Fig. 3h, both α and β modes are consistently present in the EEL signals (curves e and f) regardless of heat flow direction, indicating that the forward or reverse heat flows basically do not fundamentally change the phonon DOS. By contrast, EEG signals (curves c and d), which represent the phonon population information, show that only one mode (α or β) is distinctly dominant, directly dependent on the heat flow direction. This redistribution of interface mode populations can be considered as a unique phenomenon of non-equilibrium thermal transport, which is absent in the equilibrium state without heat flow (see Supplementary Fig. 2).

Furthermore, the AlN transverse optical (TO) phonons at 81–85 meV (overlaid curve in Fig. 3e,f) near the interface exhibit inconsistent relative intensity changes between EEG and EEL signals under forward/reverse heat flows. Although both EEL and EEG intensities decrease from bulk AlN to the interface, the EEG intensity drops ahead of EEL under forward heat flow (Fig. 3e) but lags behind EEL under reverse heat flow (Fig. 3f). The divergence between the EEL and EEG intensity

represents a change in the modal temperature occurring within about 3 nm of the interface, consistent with mode-resolved NEMD simulations[5,6] that characterize temperature variations in the AlN optical branch near the interface (Fig. 3i). Specifically, the modal temperature change is observed in the AlN TO phonons near the interface (denoted by dashed circles in Fig. 3i), which is caused by the increasing scattering channels between the AlN bulk phonons and interface modes within this spatial range, as described by the multitemperature model[3,4]. Owing to the phonon energy mismatch at the AlN–SiC interface, for which AlN TO phonons lie within the SiC bandgap (75–95 meV) as shown in Fig. 3j, most AlN bulk phonons in this energy interval become isolated modes, that is, they cannot directly propagate through the interface (see Supplementary Fig. 5) but only transfer energy through inelastic scattering including (but not limited to) interactions with the interface modes. The relative simplicity of their scattering pathways make these phonon modes ideal for studying inelastic scattering mechanisms involving interface modes. Consequently, the temperature near the interface always exhibits an earlier change on the AlN side, regardless of heat flow direction.

Figure 3i,j also reflects the non-equilibrium distribution of bulk phonons in the vicinity of the interface during thermal transport. Compared with the acoustic phonons, diffusive optical phonons exhibit a more pronounced interfacial temperature difference: on the hot side, the optical phonons maintain higher modal temperature than their acoustic counterparts, whereas this becomes opposite on the cold side. Unlike the acoustic phonons that can transport easily by means of elastic scattering or ballistic transport mechanisms, the optical phonons primarily rely on the inelastic scattering between the interface modes and bulk modes for energy exchange, thereby exhibiting larger ITR. On the basis of these, we postulate a reasonable scattering mechanism underlying the non-equilibrium distribution of interface phonons. Considering that the interface modes have different but close energy from the bulk modes, these scattering processes mainly involve two three-phonon processes that occur on the hot side and cold side of the interface, respectively. In Fig. 3j, arrows pointing from the initial states to the final states are used to illustrate the three-phonon scattering processes involving optical phonons, interface modes (α/β) and the low-energy phonon (not labelled) necessary for energy conservation. Owing to the broad momentum distributions inherent in both low-energy phonon states and non-dispersive interface modes[47], the quasi-momentum conservation condition for phonons is readily satisfied.

For a three-phonon scattering process (absorption or emission process) that is characterized by specific initial and final states, the net process follows a general principle: its probability increases with higher initial-state population and lower final-state population (see Supplementary Information Section 1). On the hot side, low-energy acoustic phonons (always serving as the initial states in absorption processes and final states in emission processes) exhibit lower modal temperatures, resulting in population below equilibrium levels. By contrast, high-energy optical phonons (functioning as the initial states in emission processes) show elevated modal temperatures and increased population relative to equilibrium. Such non-equilibrium population configurations suppress the absorption processes (represented by open green arrows) while enhancing emission processes (denoted by solid grey arrows).

On the cold side, the scenario is inverted: absorption processes (denoted by solid green arrows) are enhanced, whereas emission processes (represented by open grey arrows) are suppressed. As schematically shown in Fig. 3j, these opposing effects on scattering probability (that is, enhancement and suppression) account for the marked differences in the populations of interface modes α and β under forward and reverse heat flows. Also, this mechanism reveals further details for the phonon bridge effect[31], for which the localized interface modes with energies between the two mismatched bulk phonon energy ranges facilitate asymmetric energy transfer: energy transfer between higher-energy bulk modes and the localized interface modes occurs more readily than that between the lower-energy bulk modes and interface modes. This mechanism applies specifically to phonon modes that rely on inelastic scattering with interface modes for energy transfer. For modes exhibiting several scattering pathways, however, the scattering dynamics and heat transfer mechanisms are substantially more complex.

## Discussion and outlook

As early as 1969, G. L. Pollack posed the question of whether microscale temperature profiling across interfaces was experimentally feasible[48]. Now, using locally heated sample and a sub-nanometre electron probe, we have achieved the highest spatial resolution for temperature mapping of the heterointerface among existing experimental methods, which is fundamentally constrained by the phonon localization scale (about 2 nm) during measurement. For interface modes, the measured spatial extent is governed by the interface microstructure[31], with typical spatial broadening in both our measurements as well as previous reports[32] substantially exceeding the probe size. Building on the obtained temperature distribution, this methodology yields new insights into local thermal resistance. Given its ultra-high spatial resolution, it offers particular use for studying how the interface features (roughness and elemental mixing) and structural defects (individual dislocations, stacking faults and grain boundaries) influence local thermal conductivity.

Such experiments further enable the simultaneous characterization of ground-state phonon DOS and non-equilibrium properties of excited phonons during thermal transport. The spatial scale of inelastic scattering for bulk phonons approaching the interface (approximately 3 nm in this study) provides the critical guidance for interface engineering, such as optimizing interlayer thickness to tailor thermal properties. The theoretical framework proposed here, in which interface phonons preferentially interact with higher-energy bulk phonons irrespective of temperature gradient direction, is probably generalized to other heterojunction systems with phonon energy mismatches. This study not only clarifies the role of interface modes as 'phonon bridges'[31] but also brings a new perspective to understand ITR. Its findings hold promise for applications in advanced thermal management systems, including nanotransistors, thermoelectric materials[49] and functional devices such as thermal diodes and thermal transistors[50] for thermal rectification and thermal logic devices.

However, it should be noted that phonon transport in interface regions represents an inherently complex phenomenon, and whether the interface phonon-mediated heat flux equals the total inelastic heat flux across the interface remains an open question. Alternative scattering pathways independent of interface modes, such as phonons with MFP exceeding the interface length scale or inelastic scattering processes only confined to bulk modes near the interface, may coexist or compete with the proposed mechanisms. Furthermore, higher-order phonon processes can also play a non-negligible role in thermal conductivity, particularly in those systems with substantial mass mismatch[6]. These aspects demand thorough investigation in future studies to fully characterize the underlying transport physics.

In conclusion, we demonstrate nanoscale measurements of ITR and interface phonon transport dynamics by constructing steady-state temperature gradients and analysing the evolution of EEG/EEL signals in STEM. The spatial width of temperature drop at the interface is revealed and the ITR is estimated. Our findings reveal highly asymmetric population distributions of non-equilibrium interface phonons under forward and reverse heat flows, providing critical mechanistic insights into the underlying inelastic scattering processes. The developed methodology—enabling visualization of phonon transport through electron microscopy—presents a new framework for

investigating nanoscale heat transfer, holding great promise for advancing energy conversion technologies, information systems and next-generation thermal devices.

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

# Methods

## Hardware

The STEM-EELS data were acquired on a Nion U-HERMES200 microscope equipped with both a monochromator and aberration correctors. The EELS datasets were acquired with 60 kV beam energy and 20 mrad convergence semi-angle and 25 mrad collection semi-angle. The typical energy resolution of acquired data under these conditions was 10–13 meV. The datasets in Fig. 2c for mapping temperature from a large spatial range were obtained by using on-axis collection to obtain a higher signal-to-noise ratio. For datasets acquired in the nanoscale interface in Fig. 3, off-axis[40] collection was used to mitigate the delocalization effect owing to the dipole scattering. It has been reported that the scattering probability from interface states and anisotropy depends on the collection condition[36,51]. In our experiments, we chose a uniform off-axis setting as shown in Supplementary Fig. 2k with a 45° angle to the interface, which ensures that the scattering probabilities of the different sets of data are almost consistent and that all interface phonons with eigenvectors either perpendicular or parallel to the interface can have high excitation activity. The scanning step size is 1/6 nm and the typical spatial resolution was about 0.3 nm at the 20 mrad convergence semi-angle[32]. The typical dwell times of each pixel was 1,200–1,600 ms.

It can be seen from Fig. 1d that, at room temperature, the $\langle N \rangle$ and EEG signal is relatively weak and can be masked by the intense zero-loss peak (ZLP). Therefore, our experiments were performed at high temperature to enhance the EEG signal by increasing $\langle N \rangle$, allowing us to robustly extract the local phonon population and analyse the temperature-driven phonon transport behaviour at the interface. The heating effect of the electron beam can be ignored because the beam current for measuring vibration signals is small and the sample conducts heat well. The temperature increment is roughly[52] $\Delta T \simeq (I_b/e\kappa)(dE/dx)$, at which $I_b$ below 0.1 nA is the beam current after passing through the monochromator and $\kappa \approx 100$ W mK$^{-1}$ is the thermal conductivity of the sample. The energy loss rate per electron[53] $dE/dx \approx 0.1$ eV nm$^{-1}$. The estimated order of magnitude of $\Delta T$ is 0.1 mK, far below our measurement accuracy at present.

The in situ heating device consists of our home-made sample holder and current control system, which allows a constant current to be applied to the sample for local heating. The typical value of the total resistance of the heating circuit in the experiment is about 3–10 kΩ. After a long enough time of applying a current (generally 1–2 h), the sample becomes stable in the field of view without drifting and the temperature distribution on the sample no longer changes with time. In this case, steady-state thermal transport is achieved, that is, there is a stable temperature gradient on the sample. Then, the STEM-EELS mapping experiments were performed.

Note that it is controversial whether the nanoscale temperature in non-equilibrium state can be defined. Temperature is typically defined by statistical averages and it is commonly believed that this definition relies on local thermal equilibrium achieved at scales larger than the phonon MFP, whereas there is no temperature difference within the phonon MFP. This notion that temperature cannot be defined within the MFP is based on a premise that, at a moment, there is no time for scattering within the MFP to construct a temperature gradient. However, under steady-state thermal transport, the temperature field remains constant over time, allowing for statistical averaging over time instead of space. Over a sufficiently long period, phonon scattering can occur at any point within the MFP scale to establish a temperature gradient, although these scattering processes do not occur simultaneously. Therefore, in this case, phonon scattering can be averaged over time to determine a steady-state temperature at the scale smaller than the MFP. In fact, this is exactly how the NEMD method obtains the local temperature, which can be defined at less than 1 nm (refs. 54,55) and even atomic column resolution[56–58].

This method also has some limitations. There is no calorimetric technology integrated in the sample holder at present, making the absolute value of bulk thermal resistance unknown. Thus, only the relative thermal resistance can be obtained so far. Furthermore, the accuracy of temperature and ITR measurements needs to be further improved, which is discussed in detail in Supplementary Information Section 3.

## Sample preparation

In this study, we chose the epitaxial AlN–SiC heterojunction, which is a 2H-AlN thin film grown by metal-organic chemical vapour deposition on 4H-SiC. This is a common system in wide-bandgap semiconductor devices, with similar crystal structures and low lattice mismatch of about 1.1%. The AlN–SiC system shows minimal thermal mismatch at the interface, with bulk phonons and interface phonons DOS mostly unchanged with temperature except for a slight energy shift (see Supplementary Fig. 2). The TEM foil sample consists of AlN–SiC heterojunction on a Si support substrate with electrodes (Fusion E-chips E-FEF01-A4, Protochips). Using a scanning electron microscopy/focused ion beam (Thermo Fisher Helios G4 UX), the AlN–SiC heterojunction was welded to the notch in the middle of the supporting substrate and then processed into the shape of a 1D heat transfer path. The interface region was thinned to below 50 nm. At one end, amorphous carbon was deposited and connected to the electrode to act as a heat source. The other end was retained at a larger volume, depositing a large amount of amorphous carbon as a heat sink attached to the substrate (see Supplementary Fig. 1a). The sample thickness near the interface was controlled to be as uniform as possible during thinning so that the cross-sectional area of the heat transfer direction was approximately the same.

The sample thickness $t$ was calculated in the Gatan Digital Micrograph software from the measured plasmon peaks according to the relationship $t = \lambda \ln \frac{I_{tot}}{I_{ZLP}}$, in which $I_{tot}$ is the sum of all counts in a spectrum containing the main plasmon signal, $I_{ZLP}$ is the sum of ZLP and $\lambda$ is the electron MFP, which is determined by the convergence and collection angle, incidence energy and atomic number of the sample[59]. The detailed expression can be obtained from ref. 60. The typical thickness is about 30–40 nm in Supplementary Fig. 1b.

## EELS data processing

All acquired spectra were processed by custom-written MATLAB (R2019b) code[32]. For each dataset, EEL spectra were first registered by their cross-correlation to correct beam energy drifts. A block-matching and 3D filtering algorithm was then applied to remove Gaussian noise[61,62], in which the noise level was estimated on the basis of high-frequency elements in the Fourier domain. The same background removal method was used to obtain EEG and EEL signals. Specifically, the ZLP was removed by fitting the spectra to a Pearson function[63] in two energy windows, one before and one after the signal region. The windows were approximately 15–30 meV and 130–200 meV, which were slightly adjusted for each dataset to achieve the best fitting but remained the same for each pixel in every EELS map. This effectively removed the acoustic phonon signal below 30 meV. The temperature map was obtained by fitting the EEG and EEL signals at each pixel. For the local temperature fitting, we followed the method in the ref. 9 and fitted the temperature $T$ according to the relationship $\log(I_{loss}/I_{gain}) = \frac{1}{k_B T}\hbar\omega$. Using the fitnlm function in MATLAB, we performed a linear fit through the origin on the experimental data based on the least-squares fitting and obtain the slope $\frac{1}{k_B T}$ and error range.

Considering that the low-energy signal may be affected by the background removal, the data below about 55 meV were discarded, as we only studied the behaviour of optical phonons in this work.

For further analysis of vibration modes, Lucy–Richardson deconvolution was then used to ameliorate the broadening effect caused by the finite energy resolution, taking the ZLP obtained in the above step as the point spread function. We used five as the iteration

number, which cannot introduce artefacts to the spectral line[9]. To extract pure interface scattering features from the interface spectrum mixed with other scattering components, the difference spectrum $Diff(\omega) = S_{inter}(\omega) - a_1 S_{SiC}(\omega) - a_2 S_{AlN}(\omega)$ can be obtained by subtracting the bulk spectrum components. A similar operation has been used in the literature to extract vibration induced by the single Si atom[38] and heterointerface[32]. The process of finding $(a_1, a_2)$ was completed by the lsqcurvefit function in MATLAB based on the least-squares fitting method, which reduced $\sum_\omega (Diff(\omega))^2$ to the minimum by the loop of iterating $(a_1, a_2)$. The difference spectrum with the smallest square integral here is called the 'interface residual spectrum', which is linearly independent of the SiC and AlN spectra.

## Construction of neuroevolution potential

To build neuroevolution potential (NEP) for AlN–SiC interfaces, we first generated three datasets from density functional theory energy–force calculations containing 50 randomly displaced structures by 0.1 angstrom, 50 randomly deformed structures from −3% to 3% and 50 ab initio molecular dynamics snapshots under 900 K from 2,000 steps in each group, 450 structures in total. Each group was generated using different structure, being fully relaxed 2 × 2 × 2 AlN in the first group, fully relaxed 2 × 2 × 1 SiC in the second group and SiC–AlN interface with 2 × 2 × 2 AlN on 2 × 2 × 1 SiC in the third group. Ab initio molecular dynamics simulations were carried out using the QUICKSTEP algorithm implemented in the CP2K package[64], which makes use of the Gaussian and plane waves approaches. We used the short-range Goedecker–Teter–Hutter (GTH) double-ζ valence, single polarization short range (DZVP-MOLOPT-SR-GTH) basis set[65] and the GTH-PBEsol[66] pseudopotentials to describe the core–valence interactions. The wave-plane cut-off was set to be 500 Ry and the Brillouin zone was sampled at the Γ point. The NEPs were trained using the GPUMD package developed by Fan et al.[67], which had been proved to be capable of simulating solid interfaces[68–70]. The interaction cut-off for this potential was set to be 6 Å. The potential gave a force root means square error of 100 meV per angstrom. A parity plot showed that the atomic force and energy achieved good agreement between the ab initio benchmark and NEPs, as shown in Supplementary Fig. 3a. The rationality of the NEP function was also confirmed by the consistency of the phonon spectrum with density functional perturbation theory (DFPT) results based on the PBEsol functional, as shown in Supplementary Fig. 3b,c. Phonon DOS were also consistent with the DFPT results, as shown in Supplementary Fig. 3d. DFPT results of AlN and SiC were obtained from Materials Project databases mp-661 and mp-11714, respectively[71,72].

## Molecular dynamics simulation parameters

NEMD simulations were performed using the GPUMD package with our NEP for the target interface. For the atomic structures, the $c$ axis of wurtzite crystal is aligned with the $z$ direction. To avoid lattice mismatch in molecular dynamics simulations, we enforced the cross-sectional lattice size (along the $x$ and $y$ directions) to be identical to the lattice size of AlN. The numbers of unit cells for 2H-AlN and 4H-SiC were 12 × 12 × 72 and 12 × 12 × 36, respectively, as shown in Supplementary Fig. 4a. At the interfaces, SiC was Si-terminated and AlN was N-terminated. In molecular dynamics simulations, periodic boundary conditions were applied to all three spatial directions and a time step of 1.0 fs was used. After relaxing the structure under isobaric–isothermal conditions (NPT), isobaric–isovolumic conditions (NVT) and microcanonical ensemble (NVE) for a total of 300 ps, we froze the right boundary on the SiC side and applied a temperature difference using Langevin thermostats in the NVE for 1.2 ns. The temperatures applied on the two thermostats are ±50 K of the target mean temperature. The final temperature profile is shown in Supplementary Fig. 4b. The corresponding heat current is shown in Supplementary Fig. 4c. On the basis of NEMD simulations, the ITR was obtained by $R = (A\Delta T)/Q$,

in which $Q$ is the steady-state heat flux along the $z$ direction, $A$ is the cross-sectional area and $\Delta T$ is the temperature difference at the interface determined by extrapolating linear fits of the temperature profiles of the two sides (that is, AlN and SiC) to the interface and calculating the difference. The calculated temperature drop, ITR under different average temperature and heat transferring direction and their thermal rectification ratio are shown in Supplementary Table 1. The modal resolved temperature was defined by Feng et al.[5,6], which is shown in equations (3) and (4):

$$\dot{Q}_\lambda(t) = \frac{1}{\sqrt{N_c}} \sum_{l,b}^{N_c,n} \sqrt{m_b} \exp(-ik \cdot \mathbf{r}_{l,b}) e^*_{b,\lambda} \cdot \dot{\mathbf{u}}_{l,b;t} \tag{3}$$

$$T_\lambda = <\dot{Q}^*_\lambda(t)\dot{Q}_\lambda(t)>/k_B \tag{4}$$

$l$ and $b$ label the indexes of the primitive cells and basis atoms, with the total numbers represented by $N_c$ and $n$, respectively; $m$, $\mathbf{r}$ and $\dot{\mathbf{u}}$ are the mass, equilibrium position and velocity vector, respectively; and $e^*_{b,\lambda}$ is the complex conjugate of the eigenvector component at the basis $b$ for the mode $\lambda$. The parameters used in NEMD simulations are the same as those used in ITR calculations. Limited by memory capacity, only 8 × 8 × 8 AlN on 8 × 8 × 4 SiC interfacial systems were used in this simulation. The phonon eigenvectors and eigenvalues were calculated using the kALDo package[73] integrated with the NEP force calculator. In the calculation, the eigenvectors of AlN and SiC unit cells were mapped to both sides of the interface using dynasor package[74]. The temperature was averaged from a 5-ns NEMD trajectory with a dumping interval of 10 fs.

## Data availability

The data that support the findings of this study have been deposited in the Open Science Framework database under the accession code https://osf.io/ajt2c/.

## Code availability

The molecular-dynamics-related post-processing code and custom MATLAB codes that are used for data processing have been deposited in the Zenodo database at https://doi.org/10.5281/zenodo.15195097 (ref. 75).

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

**Acknowledgements** This work was supported by the National Natural Science Foundation of China (52125307), the National Key R&D Program of China (2019YFA0708200, 2021YFA1400502) and the '2011 Program' from the Peking-Tsinghua-IOP Collaborative Innovation Center of Quantum Matter. P.G. acknowledges support from the New Cornerstone Science Foundation through the Xplorer Prize. J.D. acknowledges support from the National Natural Science Foundation of China (92477139). We acknowledge the Electron Microscopy Laboratory of Peking University for the use of electron microscopes and the High Performance Computing Platform of Peking University for providing computational resources for the NEMD calculations. P.G. also acknowledges D.P. Yu, Z.F. Liu and E.-G. Wang for their inspiring conversations that guided this project.

**Author contributions** P.G. conceived the project. F.L. prepared the TEM sample. F.L. designed and performed the in situ STEM-EELS measurements and wrote the data-processing codes. R.M. performed NEMD calculations. F.L. performed the data analysis and figure arrangement, with the help of R.M.; Z.L. provided the heterointerface film. J.D. designed the in situ heating system under the direction of P.G.; F.L. and R.M. wrote the manuscript under the direction of P.G. All authors contributed to this work through useful discussion and/or comments on the manuscript. P.G. supervised the project.

**Competing interests** The authors declare no competing interests.

**Additional information**
**Correspondence and requests for materials** should be addressed to Peng Gao.
