## [Peer Review File · Nature]

Probing phonon transport dynamics across an interface by electron microscopy

Corresponding Author: Professor Peng Gao

Version 0:

Reviewer comments:

Referee #1

(Remarks to the Author)

Summary of the Key Results:

The manuscript presents a groundbreaking study on phonon transport at the AlN-SiC interface using in situ vibrational EELS. The authors provide high spatial resolution insights into interfacial inelastic scattering and non-equilibrium phonon behavior. They demonstrate how the direction of the temperature gradient influences heat transport and achieve sub-nanometer resolution in mapping temperature gradients. The claim that the spatial resolution of temperature is limited by phonon delocalization is particularly compelling and well-supported by experimental results.

Originality and Significance:

The work is highly original, as it combines state-of-the-art vibrational EELS techniques with novel experimental design to explore interfacial phonon dynamics at unprecedented resolution. This represents a significant advancement in nanoscale thermal transport studies.

Data & Methodology:

The approach is robust and the methodology is rigorous. The data is of high quality and is presented effectively.

Appropriate Use of Statistics and Treatment of Uncertainties:

The manuscript address statistical treatment and discuss any potential sources of error and quantify uncertainties.

Conclusions:

The conclusions are robust, valid, and reliable. Their work trully opens a new door in the invistigation of thermal properties of materials at the nanoscale, with broad implications in thermal management of nanodevices.

Suggested Improvements:

- Line 51: Cite a foundational paper or book on interface thermal resistance to strengthen the theoretical background.
- Consider language refinements and avoid superlatives. I suggest the authors to delete the words found in the following lines and simply let the readers decide if something is important, remarkable, dramatic, critical, etc.
- Line 47: "critical"
- Line 77: "vital"
- Line 81: "critical"
- Line 89: "delicately"
- Line 162: "Notably", and "remarkably"
- Line 213: "dramatic"

References:

The references cited are appropriate, and the manuscript acknowledges foundational work in the field.

Clarity and Context:

The manuscript is clearly written with also well thoughtout figures that are easy to follow. Minor language refinements as suggested above would further improve readability.

I strongly recommend the manuscript for publication in Nature after minor edits, as listed previously, are addressed by the authors.

Referee #2

(Remarks to the Author)

This paper presents high spatial resolution, modal phonon temperatures across interfaces, with results that confirm several recent, significant theoretical predictions that were not experimentally probed yet. The work not only represents a major advancement in experimental phonon spectroscopy method by achieving high spatial resolution, but also in our physical understanding of interfacial phonon transport by validating recent theoretical predictions of nonequilibrium and inelastic phonon scattering across interfaces. Such advancements have been desired by the community for quite some time. I enthusiastically support the publication of this paper, if the authors can address the following comments:

1. The explanation of interfacial phonon scattering and modal phonon energy transfer pathways is framed within the context of 3-phonon scattering. However, recent theoretical phonon spectroscopy studies (e.g., Phys. Rev. B 99, 045301 (2019)) suggest that 4-phonon or even higher-order scattering processes may play a non-negligible or even important role. The authors could expand their interpretation to include these higher-order scattering effects, while maintaining the non-equilibrium phonon population arguments.
2. Figures 2b, 2c, and 2e exhibit considerable uncertainties. A more detailed discussion on how uncertainty is managed to ensure an adequate signal-to-noise ratio would be beneficial.
3. In Figure S4(b), the label "heat current" is incorrect as the unit is in joules (J). The correct term should be "heat," as heat current should be expressed in units of W/m^2 .

Referee #3

(Remarks to the Author)

Liu et al. conducted spectrally resolved thermometry using vibrational EELS across an AlN/SiC interface under different thermal gradients. The experiments and the presentation in this manuscript are very well done and relevant to most phonon physics, including thermal transport. The measurement of interface thermal resistance at these length scales has been a long-sought goal in the thermal community, and this work not only achieves this goal but provides insights into the detailed mechanism of how heat flows across an interface and how interfacial modes mediate transport. I would recommend this article be published in Nature with a few revisions.

Concepts to address

1. The paper measures the local temperature by utilizing the gain vs loss peaks invokes the principle of detailed balance. The manuscript also makes a very important point that a (highly) non-equilibrium thermal gradient is necessary to access the thermal discontinuity, and therefore ITR, at the interface. So, there are some intricacies in these two concepts that might matter and may conflict. The principle of detailed balance is valid for a system in equilibrium as stated on line 139, e.g. a sample heated to a uniform temperature that has different EEG vs EEL intensity because of the elevated and spatially constant occupation statistics $N(\omega)$. In theory, if the measured volume and thermal gradient are small then one could invoke some sort of local equilibrium argument. In this manuscript, the locally probed volume is small however the thermal gradients are rather large, so I am not sure if "locally near equilibrium" can be invoked. I do not know when or how fast the local approximation fails or to what magnitude it impacts measured values. I wonder if the authors can comment on this concept and its implications in this manuscript?

a. That being said...even if the quantitative values are incorrect, the concepts of interfacial state occupation portrayed in the manuscript would remain valid.

b. The authors state "Notably, the scatter points align closely with the fitted line, indicating that the deviation from equilibrium state is minimal at micron scale.". Is a non-linear trend expected if the local equilibrium approximation is violated? The linear y-intercept should be zero, does "not near equilibrium" result in a y-intercept offset? I am not sure what to expect here and I think that this comment is trying to address this non-equilibrium concern but does not get the full way there.

2. In line 175-179, I am unsure what you mean by temperature limited spatial resolution. Does phonon delocalization change with temperature, or are you referring to the scattering cross-section (phonon-beam interaction) increasing with temperature? The later does not necessarily imply that delocalized interaction (impact parameter) increases, just that probability per area increases.

By the way, I quite enjoy that these measurements are not done with atomic resolution. The goal of atomic resolution has become a bit of a pragmatic goal in electron microscopy because few experiments can achieve these length scale, but for quasiparticles like phonons nm length scales are way more relevant and meaningful.

At a minimum I suggest that the authors consider rewording line 175-179 so that it is a bit clearer. Suggestion: "Considering the electron beam size of ~ 0.3 nm at a 20 mrad convergence semi-angle is much smaller than the length of temperature change, we are below temperature limited spatial resolution, determined by the degree of phonon delocalization."

3. The authors mention that they use off-axis EELS to become more sensitive to the local beam-phonon interactions.

However, they do not describe the geometry of the off-axis acquisition as described in the references below. The first reference additionally shows that the scattering probability from interface states and anisotropy depends on the collection condition, especially in materials with large anisotropy like AlN. Can the authors provide the information and a quick discussion in the text? Additionally, do you have multiple collection conditions to rule out selectivity masking some interface states?

• Eric R. Hoglund, Harrison A. Walker, Md. Kamal Hussain, De-Liang Bao, Haoyang Ni, Abdullah Mamun, Jefferey Baxter, et al. "Non-equivalent Atomic Vibrations at Interfaces in a Polar Superlattice." *Advanced Materials* 36, no. 33 (May 8, 2024): 2402925. <https://doi.org/10.1002/adma.202402925>.

• Yang, Hongbin, Yinong Zhou, Guangyao Miao, Ján Rusz, Xingxu Yan, Francisco Guzman, Xiaofeng Xu, et al. "Phonon Modes and Electron–Phonon Coupling at the FeSe/SrTiO₃ Interface." *Nature* 635, no. 8038 (November 14, 2024): 332–36. <https://doi.org/10.1038/s41586-024-08118-0>.

4. It is unclear if the suppression or enhancement of optic mode absorption vs. emission and how they depend on the modal temperature is something previously established or being established in this manuscript. I have not heard of this before, but the logic tracks from the 60-90 meV interfacial optical modes in this manuscript. Can you clarify?

5. For the interface modes there appears to be lots of details that are not addressed. I am also curious what is going on with the remaining interfacial modes outside the 60-90 meV window. Specifically, does the high-energy optical to interfacial mode always hold? Is there specific scattering or momentum conditions that are detected for different spectral regions, even though you have a non-momentum resolving (convergent) beam? Detailed thoughts below:

a. 60-90 meV bulk modes energy transfer to interfacial modes discussed the scattering appears to be between Γ and A symmetry positions with the way the two zones on each side of the interface are drawn. Can momentum energy conservation be used to say what mode is scattering too where? Is it actually A to A? Can you comment?

b. One example, there is an interesting spectral difference between forward and backward at 105 meV. The strongly dispersive behavior of these higher energy branches gives a good perspective on what modes at q and ω are "transferring" at the interface. It appears to be dominated by A to A modes.

c. The opposite seems to occur for the ~50 meV modes where AlN SiC is up hill in energy from A to Γ while for SiC AlN in this energy range no well-defined structure exists, but it looks like it is leaning toward A to Γ also. This also breaks the argument that higher energy bulk modes transfer to lower energy interfacial modes.

d. Lastly, the lowest energy ~20 meV optic modes look to be Γ (SiC) to Γ (AlN) regardless of the gradient.

6. "For the interface mode itself, the typical spatial broadening is already much larger than our beam spot size." The spatial extent of the interface mode depends on the type of interface mode. In a chemically and structurally abrupt interface, there can be modes localized precisely to atoms on the abrupt plane and there can be interfacial modes that contain atoms in both crystals vibrating within some distance from the abrupt plane. This has been demonstrated by the current authors in "Effects of localized interface phonons on heat conductivity in ingredient heterogeneous solids".

7. In line 311 you target thermal management and thermoelectric materials. Thermal management is broad reaching and directly relevant to the current measurements. The thermoelectric reference screams "I needed a connection to a material or property". This seems to have come out of nowhere and is one of many examples where thermal properties or phonon physics matter. I would suggest making this a broader reaching connection to match the scope on Nature.

8. I quite liked lines 342-354 in the methods discussing the definition of temperature, and the discussion is extremely relevant. If an abbreviated discussion could be worked into the main text that would be nice.

9. On line 43 then 175-178, the author says that chemical bonding at the interface leads to phonon scattering. Not just chemical bonding. Bonding, elemental composition, and symmetry all play a role.

10. Line 79-81: "In fact, the EEG signal is proportional to the phonon thermal occupation number reflecting changes in phonon population." Both EEG and EEL are proportional to thermal occupation. n and $n+1$.

Minor details

Line 2: "interface in an electron microscope"

Line 25 "electron energy-loss spectroscopy in an electron microscope"

Line 31: "This leads to significant changes in the modal temperature of AlN optical phonons near the interface ~3 nm within ~3 nm of the interface."

Line 32: "phonon transport dynamics at the nanoscale"

Line 42: "phonons are the primary heat carriers."

Line 45-47: "mainly arise from the localized accumulation and far-from equilibrium behavior of slow optical phonons due to ~~the phonon scattering and exacerbated by the interface~~ exacerbated phonon scattering from the interface"

Line 51: "thermal resistance (ITR), and is used to characterize"

Line 60: "At the nanoscale,"

Line 69: "energy-loss"

Lines 117-118: "population of thermally excited states $\langle N \rangle$."

Line 129-131: "~~In~~ To achieve nanoscale acquisition near the interface, we use the off axis configuration to enhance the localization nature of localized non-dipolar EEL/EEG signal as well as the space resolution of thus allowing for spatially resolved temperature maps."

Line 175-178: Consider rewording the last sentence of this paragraph. It was a bit confusing.

Line 309-310: "The ability to locally probe phonon non-equilibrium transport offers a new pathway to study the nanoscale thermal transport..."

Reviewer comments:

Referee #1

(Remarks to the Author)

In my opinion, the authors have thoroughly addressed all the comments raised by the Referees during the review process. I appreciate the authors' comprehensive discussion in their response.

The manuscript has been significantly improved, and I believe it merits publication in Nature.

I have only one minor suggestion: the authors may consider restructuring the discussion of Figure 3 into multiple paragraphs for better readability. Specifically, I recommend:

1. Line 249: Begin a new paragraph with the sentence "For a three-phonon scattering process..."
2. Line 260: Begin a new paragraph with the sentence "On the cold side..."

Beyond this, I have no further objections.

The manuscript is excellent in every aspect—from Summary and Originality to Methods and Conclusions. I believe the scientific community will embrace this work as a breakthrough, demonstrating how electron microscopy can now also be used to characterize thermal transport at the nanoscale.

Referee #2

(Remarks to the Author)

I am pleased to see that the paper has satisfactorily addressed my comments. I have also read the authors' responses to other reviewers' comments and would like to share a few additional thoughts. The paper can be further enhanced by addressing the relatively minor points outlined below, after which I enthusiastically support its publication in Nature.

1. In my view, this paper experimentally confirms phonon modal non-equilibrium and inelastic scattering, thanks to its unprecedented spatial resolution. The discussions of the underlying physical processes are generally sound. However, phonon interfacial scattering is a highly complex phenomenon, and additional transmission or scattering processes may be involved, so caution should be exercised in not prematurely ruling them out. For instance, it remains unclear from the existing literature whether the heat flux carried by interfacial modes is equivalent to the total inelastic heat flux across the interface. Furthermore, as the paper discusses, the interfacial region is typically on the order of 1-2 nm, which may be shorter than the wavelength of certain phonon modes, meaning these phonons may not "see" the interfacial region. For these reasons, it remains uncertain how many phonon modes scatter inelastically through the interfacial modes as a bridge, and how many scatter inelastically without involving the interfacial modes. The latter refers to phenomena where, in the case of three-phonon scattering, a phonon on one side directly scatters into two phonons on the other side, or, in the case of four-phonon scattering, two phonons on one side directly scatter into two new phonons on the other side. I very much like the authors' statement "This mechanism is applied to those phonon modes which require inelastic scattering through interface modes to transfer energy. However, for some modes with multiple-type scattering pathways, the scattering process and heat transfer mechanism should be much more complicated." My overall sense is that inelastic scattering processes involving interfacial phonons, as well as those not involving them, are likely co-existing and competing processes. Further studies will be needed to fully understand the role of interfacial modes. The authors have done an excellent job and can further strengthen the discussions by ensuring caution, particularly in addressing more speculative aspects, which could be clarified.

2. Previous theoretical development of modal non-equilibrium molecular dynamics (Refs. 20 and 22 in the current paper) predicted intrinsic phonon modal non-equilibrium and the existence of inelastic scattering across interfaces, but these predictions remained unvalidated for many years. One of the key contributions of the current paper is its experimental validation of these theories. To highlight this contribution of the current paper and facilitate the storyline, the statement in the abstract "...yet the nanoscale dynamics remain largely unknown due to experimental limitations in measuring the temperature of the buried interface and resolving its non-equilibrium phonon distributions^{4–7}" can be modified to "...yet the nanoscale dynamics remain largely unknown due to experimental limitations in measuring the temperature of the buried interface and resolving its non-equilibrium phonon distributions^{4–7} predicted by theories^{20,22}", or something alike. Similarly, to acknowledge prior theoretical works and better position the current work, the statement in the introduction "Nevertheless, the phonon dynamics across buried interfaces during thermal transport still remain poorly understood, leaving several important issues unresolved such as the intrinsic interface temperature drop width, the temperature gradient induced non-equilibrium phonon distribution at the interface and evolution and reversibility of interface phonon during heat transfer" can be modified to "Nevertheless, the phonon dynamics across buried interfaces during thermal transport still remain poorly understood, leaving several important predictions unvalidated such as the intrinsic interface temperature drop width,^{^refs.xx} the temperature gradient induced non-equilibrium phonon distribution at the interface,^{^refs.20,22} and evolution and reversibility of interface phonon during heat transfer^{^refs.zz}". I am quite sure that the experimental data presented in this paper will inspire further theoretical advancements, fostering a continuous loop of progress between theory and experiment.

3. In the abstract, the statement "...the interfacial inelastic scattering causes substantial non-equilibrium phonons nearby..." is confusing. In fact, scattering tends to bring the system to equilibrium instead of non-equilibrium. The mismatch of a phonon mode's bulk thermal conductivity and interfacial conductance is a key cause of phonon modal non-equilibrium. Scattering

then tends to reduce this non-equilibrium.

4. Line 228-230: "...but only transfer energy through inelastic scattering with the interface modes..." How can we rule out the possibility that a phonon scatters inelastically on its own side, creating new phonons that transmit elastically across the interface? This possibility can be acknowledged if it cannot be ruled out.

5. Line 762: "phonon model temperatures" should be corrected to "phonon modal temperatures", also in Fig. 3j it should be "modal temperature" as well?

Referee #3

(Remarks to the Author)

I am very impressed with the thoroughness of the authors' responses. I do not see any need for further edits and feel the manuscript is suited for publication.

Response to the Referee's Comments

We would like to thank the referee for the highly constructive review concerning our manuscript. In response to these comments, we have performed additional experiments and calculations, supplemented the details of experimental methodology and data processing, improved the presentation and discussion in the revised manuscript.

Revisions to the manuscript that address referees' concerns are highlighted in yellow in manuscript and shown here in red. We believe that we have addressed all the concerns made by the referees (as detailed below), and made the revision accordingly. In addition, while making necessary additions to the manuscript, in order to meet the requirements of the length of the article, we have properly adjusted and deleted some redundant expressions without changing the original meaning. A list of the major changes is attached at the end of this document.

Point to Point Responses

Reviewer #1

Comment:

Summary of the Key Results:

The manuscript presents a groundbreaking study on phonon transport at the AlN-SiC interface using in situ vibrational EELS. The authors provide high spatial resolution insights into interfacial inelastic scattering and non-equilibrium phonon behavior. They demonstrate how the direction of the temperature gradient influences heat transport and achieve sub-nanometer resolution in mapping temperature gradients. The claim that the spatial resolution of temperature is limited by phonon delocalization is particularly compelling and well-supported by experimental results.

Originality and Significance:

The work is highly original, as it combines state-of-the-art vibrational EELS techniques with novel experimental design to explore interfacial phonon dynamics at unprecedented resolution. This represents a significant advancement in nanoscale thermal transport studies.

Data & Methodology:

The approach is robust and the methodology is rigorous. The data is of high quality and is presented effectively.

Appropriate Use of Statistics and Treatment of Uncertainties:

The manuscript address statistical treatment and discuss any potential sources of error and quantify uncertainties.

Conclusions:

The conclusions are robust, valid, and reliable. Their work trully opens a new door in the invistigation of thermal properties of materials at the nanoscale, with broad implications in thermal management of nanodevices.

Response 1: We gratefully thank the referee's recognition of the innovations of our work, as well as the suggestions that helped us to further improve the manuscript.

Suggested Improvements:

1. Line 51: Cite a foundational paper or book on interface thermal resistance to strengthen the theoretical background.

Response 1.1: We thank the reviewers for their attention to the theoretical background of heat transport. We have incorporated several books and review articles on thermal transport as references to enhance the theoretical depth of the manuscript while introducing interfacial thermal resistance.

In the revised manuscript Page 3 line 57, we cite classic works in the field of thermal transport including "Thermal boundary resistance" by Swartz and Pohl [Rev. Mod. Phys. (1989). 61, 605], "Nanoscale Energy Transport and Conversion" by Gang Chen [Chen, Gang, Nanoscale Energy Transport And Conversion: A Parallel Treatment Of Electrons, Molecules, Phonons, And Photons (New York, NY, 2005; online edn, Oxford Academic, 31 Oct. 2023)], and "Interfacial thermal resistance: Past, present, and future" by Jie Chen et al. [Rev. Mod. Phys. (2022). 94, 025002] to reinforce the theoretical background.

2. Consider language refinements and avoid superlatives. I suggest the authors to delete the words found in the following lines and simply let the readers decide if something is important, remarkable, dramatic, critical, etc.

- Line 47: “critical”
- Line 77: “vital”
- Line 81: “critical”
- Line 89: “delicately”
- Line 162: “Notably”, and "remarkably”
- Line 213: “dramatic”

Response 1.2: We thank the reviewers for their corrections. We scrutinized the revised manuscript for possible exaggerations and adjusted or deleted them.

References:

The references cited are appropriate, and the manuscript acknowledges foundational work in the field.

Clarity and Context:

The manuscript is clearly written with also well thoughtout figures that are easy to follow.

Minor language refinements as suggested above would further improve readability.

I strongly recommend the manuscript for publication in Nature after minor edits, as listed previously, are addressed by the authors.

Response 1.3: We gratefully thank the reviewers’ approval of the innovations of our work, and the valuable suggestions that helped us to improve the manuscript.

Reviewer #2

Comment:

This paper presents high spatial resolution, modal phonon temperatures across interfaces, with results that confirm several recent, significant theoretical predictions that were not experimentally probed yet. The work not only represents a major advancement in experimental phonon spectroscopy method by achieving high spatial resolution, but also in our physical understanding of interfacial phonon transport by validating recent theoretical predictions of nonequilibrium and inelastic phonon scattering across interfaces. Such advancements have been desired by the community for quite some time. I enthusiastically support the publication of this paper, if the authors can address the following comments:

Response 2: We gratefully thank the reviewer for having such a deep understanding of the significance of our work.

1. The explanation of interfacial phonon scattering and modal phonon energy transfer pathways is framed within the context of 3-phonon scattering. However, recent theoretical phonon spectroscopy studies (e.g., Phys. Rev. B 99, 045301 (2019)) suggest that 4-phonon or even higher-order scattering processes may play a non-negligible or even important role. The authors could expand their interpretation to include these higher-order scattering effects, while maintaining the non-equilibrium phonon population arguments.

Response 2.1: We thank the reviewer for the constructive suggestions. We note that in Feng et al.'s theoretical work (Phys. Rev. B 99, 045301 (2019) mentioned by the reviewer), the description of four-phonon or higher order interactions was applied to interface systems with significant mass mismatch (e.g., ${}^6\text{Si}-{}^{73}\text{Ge}$ and ${}^2\text{Si}-{}^{73}\text{Ge}$ as they hypothetically proposed). In such systems, optical phonons can only facilitate thermal transport through four-phonon processes or higher-order phonon processes. However, in our SiC-AlN system, the phonon spectra exhibit substantial overlap between two materials. Additionally, previous work (Phys. Rev. X 10, 021063 (2020)) indicates that four-phonon processes contribute minimally to thermal conductivity in both AlN and SiC systems (accounting for <10% at 300K). Therefore, we believe in our case the three-

phonon scattering should be predominant and the effects of four-phonon processes or higher-order phonon processes are minimal. Notably, while our model focuses on elementary three-phonon processes governing interfacial heat transfer, the two-step scattering overall shown in Figure R4 may effectively be considered as a multi-phonon phenomenon. In the revised manuscript, we have included a brief discussion about higher-order phonon scattering contributions in Page 11 Line 272.

“Furthermore, four-phonon processes or higher-order phonon processes can also play a non-negligible role in thermal conductivity particularly in those system with significant mass mismatch [Phys. Rev. B (2019). 99, 045301].”

2. Figures 2b, 2c, and 2e exhibit considerable uncertainties. A more detailed discussion on how uncertainty is managed to ensure an adequate signal-to-noise ratio would be beneficial.

Response 2.2: The referee pointed out a technically useful discussion of the experiment. Considering the length of the main text, we added the relevant discussion in Supplementary Text 3, it reads as follows:

“The discussion in the main text addresses two types of uncertainty. The first (shown in Fig. 2b) is the uncertainty of the temperature calculation at a single data point, defined as $\pm 3\sigma$ (99.7% confidence interval) of the least squares fit result of temperature. The second uncertainty, represented by the shaded region in Fig. 2e, is the uncertainty of the average temperature obtained from multiple data points (equivalent to the error bar), derived from the standard deviation between the results of multiple acquisitions at the same position.

In our study, the uncertainty of the temperature measurement (Figures 2b, 2c, and 2e) mainly come from the quantitative ratio of EEL and EEG signals. In this case, the signal-to-noise ratio of EEG is the key to determine the uncertainty as the EEL signal is much stronger. In order to improve the signal-to-noise ratio of EEG, we have optimized experimental conditions. Firstly, the experiments were performed at relatively higher temperature. In this case, the EEG signals are stronger based-on the Bose Einstein relation. Secondly, in order to achieve high counts of EEG signal, it is certainly beneficial to

increase the integration time of the spectra acquisition. However, on the other hand, the aberration and current of electron-beam changes over time, leading to degradation of the resolution and stability of the electron-probe, which will make an additional contribution to the second type of uncertainty. So, optimization of the acquisition time is important, and it is beneficial to adopt a longer integration time under the premise that the ZLP shape is almost unchanged. In addition, if multiple acquisition and superposition can be carried out as other conditions remain unchanged, the second type of uncertainty will be reduced. Thirdly, although the higher energy resolution is better for background removal, the highest energy resolution can be only achieved at very low electron beam current, which corresponds to low signal-to-noise ratio of spectra. Again, we have optimized the energy resolution and electron counts by adjusting the EELS parameters to obtain a high energy resolution while maintaining a relatively large beam current.”

3. In Figure S4(b), the label "heat current" is incorrect as the unit is in joules (J). The correct term should be "heat," as heat current should be expressed in units of W/m².

Response 2.3: Thanks for your correction, we have corrected it in the revised Supplementary Information.

Reviewer #3

Comment:

Liu et al. conducted spectrally resolved thermometry using vibrational EELS across an AlN/SiC interface under different thermal gradients. The experiments and the presentation in this manuscript are very well done and relevant to most phonon physics, including thermal transport. The measurement of interface thermal resistance at these length scales has been a long-sought goal in the thermal community, and this work not only achieves this goal but provides insights into the detailed mechanism of how heat flows across an interface and how interfacial modes mediate transport. I would recommend this article be published in Nature with a few revisions.

Response 3: We gratefully thank the referee's recognition of the innovations of our work, as well as the comments that helped us to improve the manuscript. To further improve the manuscript, we have performed new experiments and simulations to address the referee's comments.

Suggested Improvements:

1. The paper measures the local temperature by utilizing the gain vs loss peaks invokes the principle of detailed balance. The manuscript also makes a very important point that a (highly) non-equilibrium thermal gradient is necessary to access the thermal discontinuity, and therefore ITR, at the interface. So, there are some intricacies in these two concepts that might matter and may conflict. The principle of detailed balance is valid for a system in equilibrium as stated on line 139, e.g. a sample heated to a uniform temperature that has different EEG vs EEL intensity because of the elevated and spatially constant occupation statistics $N(\omega)$. In theory, if the measured volume and thermal gradient are small then one could invoke some sort of local equilibrium argument. In this manuscript, the locally probed volume is small however the thermal gradients are rather large, so I am not sure if "locally near equilibrium" can be invoked. I do not know when or how fast the local approximation fails or to what magnitude it impacts measured values. I wonder if the authors can comment on this concept and its implications in this manuscript?

Response 3.1: Thank the referee for the valuable comment which help us to improve the clarity. The referee pointed out two concepts in the manuscript that “may conflict”: the “detailed balance” and the “non-equilibrium thermal gradient”. It should be clarified that the “detailed balance” (have been introduced in our revised manuscript now, Page 4, Line 84) here is the principle of detailed balance **for electron inelastic scattering** [Nano Lett. (2018). 18, 4556-4563], not the principle of detailed balance for phonon-phonon interactions. The actual experiment is still non-equilibrium steady-state thermal transport system. On the other hand, the temperature gradient in the bulk (except at the interface) is far from enough to invalidate the “locally near equilibrium”. In order to address the concerns of referee and potential readers, we have clarified the “detailed balance” and added the following discussion of the degree of non-equilibrium (temperature gradient) at which the “locally near equilibrium” will fail in the Supplementary Text 2, as shown in the following paragraphs:

“The “detailed balance” introduced in this work is the principle of detailed balance for high-energy electron-phonon interactions. One of the fundamental approximations of electron-phonon interaction is the “frozen lattice” approximation. One of the most important elements of this hypothesis is that the specimen thickness and the mean-free-path length for phonon excitation are both smaller than the distance travelled by the electron within the lifetime of the phonon [Acta Cryst. (1998). A54, 460-467]. That is, the time for the electron-phonon scattering process to establish equilibrium is much shorter than the average lifetime of the phonon. Thus, what the electron actually “sees” is the population of phonon in a non-equilibrium state. In previous experimental studies, some researchers have also directly introduced nonequilibrium phonon population number into the scattering cross-section formula [Nature. (2022). 606, 292-297].

Based on the above approximation, we introduce a non-equilibrium population number formulation under temperature gradients [Phys. Rev. Lett. (2018). 121, 175301], as shown in Equation (5):

$$f_{\sigma,k} = f_0(\omega_{\sigma}(k)) - \tau v_{\sigma,i}(k) \frac{\partial f_0}{\partial T} \frac{\partial T}{\partial x_i} \quad (5)$$

where f_0 is the Bose-Einstein distribution function at equilibrium, τ is the relaxation

time of the phonon, and v is the phonon group velocity at the corresponding momentum point.

Considering the processes of electron and phonon scattering in materials, the principle of detailed balancing rule requires:

$$\frac{P_{\text{loss}}}{P_{\text{gain}}} = \frac{I_{\text{loss}}}{I_{\text{gain}}} = \frac{\langle n \rangle + 1}{\langle n \rangle} \quad (6)$$

where $\langle n \rangle$ is the Bosonic distribution under equilibrium states. Then the ratio of the electron energy loss to the electron energy gain spectral intensity in the equilibrium state satisfies:

$$\frac{I_{\text{loss}}}{I_{\text{gain}}} = \exp\left(\frac{\hbar\omega}{k_B T}\right) \quad (7)$$

Substituting Equation (5) into Equation (6), we obtain Equation (8):

$$\frac{I_{\text{loss}}}{I_{\text{gain}}} = \frac{\beta - \sigma}{\frac{\beta}{\beta + 1} - \sigma} \quad (8)$$

where $\beta = \exp\left(\frac{\hbar\omega}{k_B T}\right) - 1$, $\sigma = \tau v_i \frac{\partial T}{\partial x_i} \frac{\hbar\omega}{k_B T^2}$. While $\frac{\partial T}{\partial x_i} = 0$, $\sigma = 0$, the system is at equilibrium state, then the equation degenerates to Equation (7).

We have solved this model exactly numerically. For wurtzite-AlN optical phonons, the mean free path of TO phonons at 300 K is approximately 1 nm [Diam. Relat. Mater. (2007). 16, 1413–1416], and the mean free path can be approximated as the product of the phonon relaxation time τ and the phonon group velocity v . The typical temperature gradient in the non-interface regions of the heated sample is 0.18 K/nm. We selected temperature gradients of 0 K/nm, 0.18 K/nm (temperature gradient in bulk measured by our experiments), 5 K/nm, 20 K/nm, and 60 K/nm to plot the relationship between $\log(I_{\text{loss}}/I_{\text{gain}})$ and ω , as shown in Figure S6. It can be seen from Figure S6 that the non-equilibrium $\log(I_{\text{loss}}/I_{\text{gain}})$ curves still pass through the origin, but produce a nonlinear trend and a slope change than the equilibrium curve at high energy region. Within 5K /nm (orange line), the temperature gradient only slightly affects the slope of the curve, which is basically within the experimental error range.

In this work, we measured the temperature map at the micrometer scale, and all the

measured temperature gradients in bulk did not exceed 0.18 K/nm (red line), which is almost no difference from the temperature fitting in the equilibrium state, indicating the applicability of the local near equilibrium approximation. While the temperature drop near the interface occurs in the range of $\sim 2\text{nm}$, and the degree of non-equilibrium is two to three orders of magnitude higher than in bulk, enough to cause significant non-equilibrium effects. Considering the complexity of defining temperature under high non-equilibrium degree at interface, we do not focus on the exact value of temperature in the related discussion in Figure 3.”

Figure R1(Revised Figure S6). Function of $\log(I_{\text{loss}}/I_{\text{gain}})$ versus ω under temperature gradient induced non-equilibrium state. Blue line ($\nabla T=0$ K/nm) and orange line ($\nabla T=0.18$ K/nm) are too close to be distinguished.

We have modified the corresponding description of the main text to resolve possible conflicts and it reads as follows (Page 6 Line 147):

“where k_B is the Boltzmann factor, ω is phonon frequency. Theoretically the fast electron-phonon interaction time is much smaller than the relaxation time of the phonon[Acta Cryst. (1998). A54, 460-467], so the equation (1) always holds, which is also required by the PDB [Nano Lett. (2018). 18, 4556–4563]. Under the temperature gradient, the phonon population n deviates from $\langle N \rangle$. However, this small deviation in our experiment can be negligible in the temperature calculation, i.e., the equation (2) is still valid (see Supplementary Text 2 for a detailed discussion).”

a. That being said...even if the quantitative values are incorrect, the concepts of interfacial state occupation portrayed in the manuscript would remain valid.

Response 3.1a: This statement is entirely correct, and our original manuscript has avoided the possible contradiction. It is known from Response 3.1 above that under the experimental temperature gradient in the bulk, the deviation between the slope of equilibrium temperature fitting and the actual (non-equilibrium) temperature is negligible. In the vicinity of the interface, considering the complexity of defining temperature under high non-equilibrium degree at interface, we do not focus on the exact value of temperature, and instead we use the intensity of EEG spectrum to represent the variation of phonon population near the interface. In other words, “the concepts of interfacial state occupation portrayed” does not depend on a precise definition of the interface temperature.

b. The authors state “Notably, the scatter points align closely with the fitted line, indicating that the deviation from equilibrium state is minimal at micron scale.”. Is a non-linear trend expected if the local equilibrium approximation is violated? The linear y-intercept should be zero, does “not near equilibrium” result in a y-intercept offset? I am not sure what to expect here and I think that this comment is trying to address this non-equilibrium concern but does not get the full way there.

Response 3.1b: We have shown in Response 3.1 by numerical analysis that if the local equilibrium approximation is violated (at a large temperature gradient), a nonlinear trend does occur, as shown by the green line in **Figure R1**. However, “the scatter points align closely with the fitted line” show that there is no obvious nonlinear behavior caused by

non-equilibrium in the experimental data. We have also shown in Response 3.1 that the degree of non-equilibrium under our experimental conditions does not actually cause nonlinear trend, and there is no obvious difference from the fitting experimental results under the equilibrium state, which proves the applicability of the local equilibrium approximation.

2. In line 175-179, I am unsure what you mean by temperature limited spatial resolution. Does phonon delocalization change with temperature, or are you referring to the scattering cross-section (phonon-beam interaction) increasing with temperature? The later does not necessarily imply that delocalized interaction (impact parameter) increases, just that probability per area increases.

By the way, I quite enjoy that these measurements are not done with atomic resolution. The goal of atomic resolution has become a bit of a pragmatic goal in electron microscopy because few experiments can achieve these length scale, but for quasiparticles like phonons nm length scales are way more relevant and meaningful.

At a minimum I suggest that the authors consider rewording line 175-179 so that it is a bit clearer. Suggestion: "Considering the electron beam size of ~ 0.3 nm at a 20 mrad convergence semi-angle is much smaller than the length of temperature change, we are below temperature limited spatial resolution, determined by the degree of phonon delocalization."

Response 3.2: We apologize for any ambiguity in the expression of our original sentence. We didn't mean to talk about the relationship between phonon delocalization and temperature. What we intended to mean was that the **spatial resolution of the temperature map** is determined by the phonon delocalization size, and the spatial resolution of the instrument (spot size) is lower than this size. The "delocalization" here refers to the spatial characteristic length of phonon spectrum changing. We very much agree with you that "for quasiparticles like phonons nm length scales are way more relevant and meaningful." What we want to show is that the ~ 0.3 nm beam size is enough to spatially distinguish phonon variations.

According to the referee's comment, we have made the following modifications to the

revised manuscript Page 7 Line 183 to avoid the appearance of this word.

“Considering the electron beam size of ~0.3 nm at a 20 mrad convergence semi-angle is much smaller than the length of temperature change, we are below the spatial resolution of temperature map, limited by the spatial characteristic length of phonon spectrum changing.”

3. The authors mention that they use off-axis EELS to become more sensitive to the local beam- phonon interactions. However, they do not describe the geometry of the off-axis acquisition as described in the references below. The first reference additionally shows that the scattering probability from interface states and anisotropy depends on the collection condition, especially in materials with large anisotropy like AlN. Can the authors provide the information and a quick discussion in the text? Additionally, do you have multiple collection conditions to rule out selectivity masking some interface states?

- Eric R. Hoglund, Harrison A. Walker, Md. Kamal Hussain, De - Liang Bao, Haoyang Ni, Abdullah Mamun, Jefferey Baxter, et al. “Non-equivalent Atomic Vibrations at Interfaces in a Polar Superlattice.” *Advanced Materials* 36, no. 33 (May 8, 2024): 2402925. <https://doi.org/10.1002/adma.202402925>.
- Yang, Hongbin, Yinong Zhou, Guangyao Miao, Ján Ruzs, Xingxu Yan, Francisco Guzman, Xiaofeng Xu, et al. “Phonon Modes and Electron–Phonon Coupling at the FeSe/SrTiO₃ Interface.” *Nature* 635, no. 8038 (November 14, 2024): 332–36. <https://doi.org/10.1038/s41586-024-08118-0>.

Response 3.3: We thank the referee for suggestions regarding the description of experimental details. In our experiments, we adopted the same off-axis condition for each sample, avoiding the difference of spectral lines caused by the scattering cross section. Since the orientation of the sample placed in the sample stage is fixed (as shown in Figure S1a), the angle between the interface direction and the EELS detector is fixed (~45° in our device), so we can ensure that the off-axis direction is consistent with the interface direction for each experiment. The off-axis direction we used in the experiment is shown in **Figure. R2**, which is also added into the Supplementary Information as Figure. S2k.

Figure. R2 (Revised Figure S2k) The off-axis direction of the experiment. The red circle and green circles are transmission disk and diffraction disks respectively. The yellow disk represents the EELS entrance aperture, with its center oriented at a 45° angle to the interface and its outer edge precisely tangent to the transmission spot.

In fact, there are numerous advantages to using this off-axis condition. The formula for the scattering cross section of energetic electrons and phonons [Ultramicroscopy (2023). 253,113818] is shown in equation (1)

$$\frac{d^2\sigma}{d\omega d\Omega} \propto \sum_{mode \lambda} |F_\lambda(\mathbf{q})|^2 \left[\frac{n_q + 1}{\omega_\lambda(\mathbf{q})} \delta(\omega - \omega_\lambda(\mathbf{q})) + \frac{n_q}{\omega_\lambda(\mathbf{q})} \delta(\omega + \omega_\lambda(\mathbf{q})) \right] \quad (1)$$

$$F_\lambda(\mathbf{q}) \propto \frac{1}{q^2} \sum_{atom k} \frac{1}{\sqrt{M_k}} e^{-i\mathbf{q} \cdot \mathbf{r}_k} e^{-W_k(\mathbf{q})} Z_k[\mathbf{e}_\lambda(k, \mathbf{q}) \cdot \mathbf{q}]$$

This off-axis condition, combined with the use of a large convergence semi-angle, allows the product of the momentum transfer direction and the vibrational eigenvector (i.e., the $\mathbf{e}_\lambda(k, \mathbf{q}) \cdot \mathbf{q}$ term) to be nonzero for almost all interface modes, regardless of whether the vibrational modes are parallel or perpendicular to the interface.

Additionally, it is true that changing the off-axis conditions can affect the modes that the electron beam can excite. However, it can be seen in equation (1) that the proportion of scattering cross-section on energy gain side and energy loss side can only be modulated by the population number $\langle n \rangle$. The change of $F(\lambda)$ caused by the changing of scattering activity or scattering form factor cannot change the relative intensity ratio of energy gain to energy loss. This also means that although there is a strong anisotropy in the scattering, this anisotropy has no effect on the measurement of the population number, i.e. on the measurement of I_{loss}/I_{gain} and the temperature.

In revised manuscript Page 13 Line 338, we added the description and a brief discussion about our experimental settings:

“It has been reported that the scattering probability from interface states and anisotropy depends on the collection condition [Adv. Mater. (2024). 36, 2402925; Nature (2024). 635, 332-336]. In our experiments, we chose a uniform off-axis setting as shown in Fig. S2k with a 45° angle to the interface, which ensures that the scattering probabilities of the different sets of data are almost consistent, and that all interface phonons with eigenvectors either perpendicular or parallel to the interface can have high excitation activity.”

4. It is unclear if the suppression or enhancement of optic mode absorption vs. emission and how they depend on the modal temperature is something previously established or being established in this manuscript. I have not heard of this before, but the logic tracks from the 60-90 meV interfacial optical modes in this manuscript. Can you clarify?

Response 3.4: The explanation of the mechanism behind our experimental phenomenon here is proposed by us for the first time, but we do base it on some existing theories. So, the relevant theory should be said as “being established”.

Indeed, interface phonon-dominated optical branch emission and absorption during heat transport has been explored in preliminary studies. For example, J. Maassen et al. studied heat transfer near the Si-Ge interface using the McKelvey-Shockley flux method, focusing on size effects [APL Mater. (2019). 7, 013203]. They observed heat flow around 48 meV at the interface, transferring energy to mid-frequency modes. Similarly, Y. Guo et al. used the inelastic atomic-Green's function method to identify spectral heat transfer from optical modes to interface modes in Si-Ge system [Phys. Rev. B (2021). 103, 174306]. However, neither work addresses what will happen under reverse heat flow. Our work provides the first experimental evidence for that the role of the interface mode changes under both forward and reverse heat flow conditions.

The main phenomenon observed in our experiments is the enhancement and weakening of the EEG signal of the interfacial local mode near the interface. The two interface modes located in the SiC optical bandgap - the mode near 70 meV and the mode

near 90 meV - are significantly different under forward and reverse heat flow conditions. The mode around 90 meV is stronger during heat transfer from SiC to AlN, while the mode around 70 meV is stronger for reverse heat transfer. The existing theoretical models are not sufficient for the mechanism proposed in our work. The model proposed here is similar to Le Chatelier's principle in chemistry, i.e., a system that deviates from its equilibrium state always tends to return to its equilibrium state. We decompose the three-phonon scattering process into two parts, each corresponding to a phonon generation or annihilation process occurring on one side of the interface. Among them, phonons at the hot end with higher modal temperatures (or phonons at the cold end with lower modal temperatures) have a greater degree of non-equilibrium, thus the scattering processes corresponding to these phonons are more likely to occur. The interface modes act as a "transit station" for the two processes on either side of the interface. The interface modes located on the upper and lower sides of the bandgap play different roles in this process. A simplified schematic is shown in **Figure R3**.

Figure. R3 Schematic illustration of the three-phonon process across the interface with the participation of interface modes (taking heat transfer from AlN to SiC as an example). The green arrow indicates the absorption process (annihilation of two phonons produces one phonon), while the gray arrow indicates the emission process (annihilation of one phonon produces two phonons). The black arrow (bottom) means that low-energy acoustic phonons have high transmittance and low modal ITR due to energy matching.

5. For the interface modes there appears to be lots of details that are not addressed. I am also curious what is going on with the remaining interfacial modes outside the 60-90 meV window. Specifically, does the high-energy optical to interfacial mode always hold? Is there specific scattering or momentum conditions that are detected for different spectral regions, even though you have a non-momentum resolving (convergent) beam?

Response 3.5: In fact, phonon scattering occurring at the interface is actually complex, often involving multiple phonons in multiple steps [Phys. Rev. B (2022). 106, 195435]. In our manuscript, what we focus is pure inelastic scattering. In this case, the **excited** interface mode, as a temporary state, acts as a transfer station for the phonons from the two sides with mismatched energy and momentum. For the other cases, the situation is more complicated, as described in Response 3.5b-d.

Here we consider for the inelastic scattering, for simplicity, a two-step process, each of which is a three-phonon process involving the generation or annihilation of an interface mode. Specifically, as shown in Figure R3, the step-1 is the annihilation of a high-energy (AlN-TO) phonon and the generation of an interface mode, accompanied by the production or annihilation of a low-energy phonon. The step-2 annihilates the interface mode generated in the step-1, producing high-energy (SiC-TO/LO) phonons on the other side, accompanied by the production or annihilation of a low-energy phonon. The energy difference between the interface mode and the high-energy phonons is very small (within ~ 20 meV), so the low-energy phonons produced or annihilated in the two steps have energies of 0-20 meV. Since these low-energy phonons can have a wide momentum distribution, and interface modes do not propagate as traveling waves, then these modes can be viewed non-dispersive [Sci. Rep. (2017). 7, 11011; Langmuir (2024). 40, 19, 10008–10023] on the direction perpendicular to the interface, meaning that the conservation of quasi-momentum for phonons in this region is easily satisfied. Furthermore, since the momentum of the two low-energy phonons involved in these two steps can be different, the two sides of the first and final states of the whole process do not need to have the same momentum. Under such circumstances, energy conservation becomes the core focus of the discussion.

Figure R4 a. Phonon dispersion of SiC and AlN along high-symmetry path Γ -M-K- Γ in Fig. S3. **b.** Revised schematic illustration in Fig. 3j.

For this reason, we deliberately avoided expressions such as "scattering from the Γ point to the A point" to ensure rigor in the discussion. This premise also explains the appropriateness of using the converged beam condition, as our primary focus is on scattering conditions in the energy dimension, without delving too much into momentum transfer. The dispersion relationship we show along the Γ -A direction in the paper is mainly to intuitively demonstrate that different phonons exhibit significantly non-equilibrium temperature distributions along the heat transfer direction. Note that the SiC-gap and phonon mismatches of 75-95 meV are present throughout the Brillouin zone, as shown in Figure R4a. To avoid possible misunderstandings, we modified Figure 3j, as shown in Figure R4b.

Detailed thoughts below:

- a. 60-90 meV bulk modes energy transfer to interfacial modes discussed the scattering appears to be between Γ and A symmetry positions with the way the two zones on each side of the interface are drawn. Can momentum energy conservation be used to say what mode is scattering to where? Is it actually A to A? Can you comment?

Response 3.5a: We sincerely apologize for the confusion caused by the formatting of the figures in the article. It appears to suggest a transition from a high-symmetry point Γ to A high-symmetry point A. However, in our discussion, no information regarding momentum space is included. The scattering processes were described in details in Figure

R3 and Response 3.5.

To avoid any misunderstanding, we have provided explanatory notes in the figure caption on revised Figure 3, and also discussed the non-dispersive nature of interface modes in the heat transfer direction on Page 10 Line 244 of the revised text. We also modified Figure. 3j which contains the entire Brillouin zone on Γ -A direction, as shown in Figure R4b.

“We use arrows pointing from the initial states to the final states in Fig. 3j to represent the three-phonon scattering processes associated with optical phonons, interface modes (α/β) and the low-energy phonon (not labeled) required for energy conservation. Since both of the low-energy phonon states and non-dispersive interface modes [Sci. Rep. (2017). 7, 11011; Langmuir (2024). 40, 10008–10023] can have a wide momentum distribution, the conservation of quasi-momentum for phonons is easily satisfied.”

- b. One example, there is an interesting spectral difference between forward and backward at 105 meV. The strongly dispersive behavior of these higher energy branches gives a good perspective on what modes at q and ω are “transferring” at the interface. It appears to be dominated by A to A modes.
- c. The opposite seems to occur for the ~ 50 meV modes where $\text{AlN} \rightarrow \text{SiC}$ is up hill in energy from A to Γ while for $\text{SiC} \rightarrow \text{AlN}$ in this energy range no well-defined structure exists, but it looks like it is leaning toward A to Γ also. This also breaks the argument that higher energy bulk modes transfer to lower energy interfacial modes.
- d. Lastly, the lowest energy ~ 20 meV optic modes look to be Γ (SiC) to Γ (AlN) regardless of the gradient.

Response 3.5b-d: The modes in the 60-90 meV range in AlN have a relatively single scattering path and are easier to be discussed because they are isolated modes (mentioned in revised manuscript Page 9 Line 227) located within the phonon bandgap of SiC and can only be transmitted across the interface by inelastic scattering with the interface modes.

However, other modes beyond this energy range have multiple scattering paths (including inelastic and elastic scatterings), which makes it difficult for us to fully

describe their scattering behavior by this simple model. For example, for other interface localized modes that are not located in the phonon bandgap (such as those mentioned by the referee 105 meV, ~50 meV, and ~20 meV), the heat transfer can be competitively contributed by localized mode inelastic scattering and elastic scattering or even ballistic scattering of other delocalized modes, which unfortunately, is impossible to decouple for the current experiments. To better address their transmission mechanisms in future, a more complicated and specialized calculations are needed.

In the revised manuscript Page 10 Line 269, we further clarified the scope of applicability of this mechanism.

“This mechanism is applied to those phonon modes which require inelastic scattering through interface modes to transfer energy. However, for some modes with multiple-type scattering pathways, the scattering process and heat transfer mechanism should be much more complicated. Furthermore, four-phonon processes or higher-order phonon processes can also play a non-negligible role in thermal conductivity particularly in those systems with significant mass-mismatch [Phys. Rev. B (2019). 95, 195202]. These circumstances needs to be delved deeper by further work.”

6. “For the interface mode itself, the typical spatial broadening is already much larger than our beam spot size.” The spatial extent of the interface mode depends on the type of interface mode. In a chemically and structurally abrupt interface, there can be modes localized precisely to atoms on the abrupt plane and there can be interfacial modes that contain atoms in both crystals vibrating within some distance from the abrupt plane. This has been demonstrated by the current authors in “Effects of localized interface phonons on heat conductivity in ingredient heterogeneous solids”.

Response 3.6: We agree with the referee that the spatial extent of the interface mode depends on the type of interfaces. Generally, the measured spatial broadening of localized interface modes was about 1.5 nm-1.8 nm [Chin. Phys. Lett. (2023). 40, 036801; Nature. (2021). 599, 399-403; Adv. Mater. (2024). 2402925, Nature. Comm. (2021). 12, 6901] even for atomically sharp interfaces. In our study, the broadening is close to that reported in literatures. In other words, the broadening in both of these reported literatures and our

study has already been significantly larger than the spot size. We have added a quick discussion in Page 11 Line 284 of the modified manuscript:

“For the interface mode itself, the measured spatial extent depends on the interface microstructure [Chin. Phys. Lett. (2023). 40, 036801]. The typical spatial broadening in previous reports [Nature (2021). 599, 399–403] and our work is already much larger than our probe size.”

7. In line 311 you target thermal management and thermoelectric materials. Thermal management is broad reaching and directly relevant to the current measurements. The thermoelectric reference screams "I needed a connection to a material or property". This seems to have come out of nowhere and is one of many examples where thermal properties or phonon physics matter. I would suggest making this a broader reaching connection to match the scope on Nature.

Response 3.7: Thank you for your valuable comment. We modify this sentence in Page 12 Line 322 to match the scope on Nature, it read as follows:

“The ability to locally probe phonon non-equilibrium transport helps to link the thermal properties of materials to phonon physics, providing a new pathway to study the nanoscale thermal transport in thermal management materials, and enabling the phonon engineering towards desired thermal properties, which is particularly useful for today's energy conversion and information technologies.”

8. I quite liked lines 342-354 in the methods discussing the definition of temperature, and the discussion is extremely relevant. If an **abbreviated discussion** could be worked into the main text that would be nice.

Response 3.8: Thanks to your suggestion, we have inserted this discussion into the text Page 6 Line 131, it read as follow:

“When the non-equilibrium system relaxed to steady state, the local temperature can be defined as time-average at nanoscale that smaller than the mean free path (MFP)

of phonon (see Methods for detailed discussion).”

9. On line 43 then 175-178, the author says that chemical bonding at the interface leads to phonon scattering. Not just chemical bonding. Bonding, elemental composition, and symmetry all play a role.

Response 3.9: Thanks to the referee for the reminder and we have added your suggestion in revised manuscript Page 9 Line 47.

“At the interface, mismatches in phonon energy and momentum due to discontinuities in chemical bonding [Int. J. Heat. Mass. Transf. (2024). 232, 125943], elemental composition [Nature. Comm. (2021). 12, 6901] and symmetry lead to substantial phonon scattering, thus increasing thermal resistance and intensifying the hotspots.”

10. Line 79-81: “In fact, the EEG signal is proportional to the phonon thermal occupation number reflecting changes in phonon population.” Both EEG and EEL are proportional to thermal occupation. n and $n+1$.

Response 3.10: Thanks to the referee for the reminder of what kind of principle the specific values of signal strength follow. We should note that in most cases (non-extremely high temperature, non-extremely low frequency) n is small relative to 1, so considering the relative change in signal strength ($\frac{\Delta n}{n} > \frac{\Delta n}{n+1}$), the EEG signal can more clearly reflect the relative change of phonon population. We added a quick discussion in revised manuscript Page 5 Line 118.

“The EEG signal intensity directly reflects the population of thermally excited states $\langle N \rangle$ (Fig. 1d), while EEL signal intensity represents the total number of excited and ground state phonons $1 + \langle N \rangle$. EEG signals better reflect the relative change of phonon population ($\frac{\Delta \langle N \rangle}{\langle N \rangle} > \frac{\Delta \langle N \rangle}{\langle N \rangle + 1}$), particularly for $\langle N \rangle \ll 1$ (non-high temperature).”

Minor details

Line 2: “interface in an electron microscope”

Line 25 “electron energy-loss spectroscopy in an electron microscope”

Line 31: “This leads to significant changes in the modal temperature of AlN optical phonons ~~near~~
~~the interface ~3 nm~~ within ~3 nm of the interface.”

Line 32: “phonon transport dynamics at the
nanoscale” Line 42: “phonons are the
primary heat carriers.”

Line 45-47: “mainly arise from the localized accumulation and far-from equilibrium behavior of slow optical phonons due to ~~the phonon scattering and exacerbated by the interface~~
~~exacerbated~~
phonon scattering from the interface”

Line 51: “thermal resistance (ITR), and is used to characterize”

Line 60: “At the
nanoscale,” Line

69: “energy-
loss”

Lines 117-118: “population of thermally excited states $\langle N \rangle$.”

Line 129-131: “~~In~~To achieve nanoscale acquisition near the interface, we use the off axis configuration to enhance the ~~localization nature of~~ localized non-dipolar EEL/EEG signal ~~as well~~
~~as the space resolution of~~ thus allowing for spatially resolved temperature maps.”

Line 175-178: Consider rewording the last sentence of this paragraph. It was a bit confusing.

Line 309-310: “The ability to locally probe phonon non-equilibrium transport offers a new
pathway to study the nanoscale thermal transport...”

Response 3.11: We greatly thank the reviewers for their meticulous corrections, and we have carefully checked these details and made corrections in the revised manuscript.

Major change list

We fully accept the specific suggestions for text in “minor details” of referee#3, which are not shown here.

1. Page 3, Line 48, “elemental composition¹⁴ and symmetry” has been added as suggested by referee#3.
2. Page 3, Line 50, the text has been revised to be: “Especially in the transistor drain region, nanoscale hotspots¹⁵ originate from...” as requested by the referee#1 to delete “notably”.
3. Page 3, Line 52, the text has been revised to be: “Therefore, studying the non-equilibrium phonon transport at interfaces is necessary.” as requested by the referee#1 to replace “critical”.
4. Page 3, Line 57, “which has received a lot of attention since the last century¹⁷⁻¹⁹” has been added as suggested by referee#1.
5. Page 3, Line 59, the text has been revised to be: “the study of ITR encompasses experimental methods such as time-/frequency-domain thermal reflectance (TDTR^{24,25}/FDTR²⁶).” to reduce the number of words without changing the original meaning.
6. Page 3, Line 67, the text has been revised to be: “and electron self-heating in scanning electron microscope achieves thermal resistance mapping at ~20 nm resolution³⁰.” to make the expression unambiguous.
7. Page 4, Line 70, the text has been revised to be: “still remain poorly understood” to make the expression unambiguous.
8. Page 4, Line 78, the text has been revised to be: “providing important insights into ITR. However, investigating non-equilibrium phonon transport behavior across the interface necessitates establishing large temperature gradients at interfaces during STEM-EELS measurements.” as requested by the referee#1 to delete “vital” and to reduce the number of words without changing the original meaning.
9. Page 4, Line 81, the text has been revised to be: “Moreover, it requires the information of phonon populations, which deviate from the Bose-Einstein distribution in non-equilibrium states. Phonon populations can be directly reflected in electron energy-loss/-gain (EEL/EEG) signals by the principle of detailed balancing (PDB), and can be used for nanoscale temperature measurements⁴⁻⁶.” to address referee#3's concerns.
10. Page 4, Line 92, the text has been revised to be: “A substantial and stable ~180 K/ μ m temperature gradient is generated at a thin foil heterointerface for STEM-EELS characterization.” as requested by the referee#1 to delete “delicately”.
11. Page 4, Line 98, the text has been revised to be: “Excited phonon states analysis from the EEG signals shows the interface scattering leads to substantial non-equilibrium phonons within ~3 nm near the interface, further altering the nearby AlN optical phonon modal temperature.” to reduce the number of words without changing the original meaning.
12. Page 5, Line 108, the text has been revised to be: “in-situ STEM-EELS approach with steady-state heat flow” to reduce the number of words without changing the original meaning.

13. Page 5, Line 119, “thermally” has been added.
14. Page 5, Line 120, “EEG signals better reflect the relative change of phonon population ($\frac{\Delta\langle N \rangle}{\langle N \rangle} > \frac{\Delta\langle N \rangle}{\langle N \rangle + 1}$), particularly for $\langle N \rangle \ll 1$ (non-high temperature)” has been added to address referee#3's concerns.
15. Page 6, Line 131, “When the non-equilibrium system relaxed to steady state, the local temperature can be defined as time-average at nanoscale that smaller than the mean free path (MFP) of phonon (see Methods for detailed discussion).” has been added to address referee#3's concerns.
16. Page 6, Line 147, the text has been revised to be: “Theoretically the fast electron-phonon interaction time is much smaller than the relaxation time of the phonon⁴³, so the equation (1) always holds, which is also required by the PDB⁵. Under the temperature gradient, the phonon population n deviates from $\langle N \rangle$. However, this small deviation in our experiment can be negligible in the temperature calculation, i.e., the equation (2) is still valid (see Supplementary Text 2 for a detailed discussion).” to address referee#3's concerns.
17. Page 8, Line 185, the text has been revised to be: “we are below the spatial resolution of temperature map, limited by the spatial characteristic length of phonon spectrum changing.” to address referee#3's concerns and make the expression unambiguous.
18. Page 9, Line 228, the text has been revised to be: “cannot directly propagate through the interface (see Fig. S5), but only transfer energy through inelastic scattering with the interface modes. These isolated modes with a single scattering path are suitable for studying the inelastic scattering mechanism involving interface modes.” to address referee#3's concerns.
19. Page 10, Line 246, the text has been revised to be: “associated with optical phonons, interface modes (α/β) and the low-energy phonon (not labeled) required for energy conservation. Since both of the low-energy phonon states and non-dispersive interface modes^{57,58} can have a wide momentum distribution, the conservation of quasi-momentum for phonons is easily satisfied.” to address referee#3's concerns and make our statements clearer.
20. Page 10, Line 269, “This mechanism is applied to those phonon modes which require inelastic scattering through interface modes to transfer energy. However, for some modes with multiple-type scattering pathways, the scattering process and heat transfer mechanism should be much more complicated. Furthermore, four-phonon processes or higher-order phonon processes can also play a non-negligible role in thermal conductivity particularly in those systems with significant mass-mismatch²². These circumstances needs to be delved deeper by further work.” has been added to address referee#2 and referee#3's concerns.
21. Page 11, Line 279, the text has been revised to be: “Now using locally heated sample with ~ 0.3 nm electron-probe, we have observed 10-20 K temperature drop across the AlN-SiC interface within ~ 2 nm spatial scales, achieving the highest spatial resolution among the existing experimental methods.” as requested by the referee#1 to delete “delicately and to reduce the number of words without changing the original meaning.

22. Page 11, Line 284, the text has been revised to be: “For the interface mode itself, the measured spatial extent depends on the interface microstructure³¹. The typical spatial broadening in previous reports³² and our work is already much larger than our probe size.” to address referee#3's concerns.
23. Page 11, Line 290, the text has been revised to be: “it can directly evaluate the effect of interfacial roughness and elemental mixing on thermal conductivity, and characterize the thermal resistance at individual dislocations, stacking faults, and grain boundaries.” to reduce the number of words without changing the original meaning.
24. Page 12, Line 301, the text has been revised to be: “Comparing the forward/reverse heat flow reveals distinct non-equilibrium behaviors: interface phonons prefer to interact with phonons of higher-energies at both the hot and cold ends. This theory can also be generalized to other heterojunction systems with phonon mismatches.” to reduce the number of words without changing the original meaning.
25. Page 12, Line 313, the text has been revised to be: “Additionally, the accuracy of temperature and ITR measurements needs to be further improved, which is discussed in detail in Supplementary Text 3.” to address referee#2's concerns.
26. Page 12, Line 323, the text has been revised to be: “helps to link the thermal properties of materials to phonon physics, providing a new pathway to study the nanoscale thermal transport in thermal management materials, and enabling the phonon engineering towards desired thermal properties” to address referee#3's concerns.
27. Page 13, Line 338, “It has been reported that the scattering probability from interface states and anisotropy depends on the collection condition^{37,65}. In our experiments, we chose a uniform off-axis setting as shown in Fig. S2k with a 45° angle to the interface, which ensures that the scattering probabilities of the different sets of data are almost consistent, and that all interface phonons with eigenvectors either perpendicular or parallel to the interface can have high excitation activity.” as well as Figure S2k has been added to address referee#3's concerns.
28. Page 14, Line 373, “In fact, this is exactly how NEMD method obtains the local temperature, which can be defined less than 1 nm^{68,69} and even atomic column resolution^{70–72}.” has been added to strengthen the argument.
29. The diagram in Figure 3j has been revised for better visualization and to avoid ambiguity to address referee#3's concerns.
30. In Supplementary Information, “Supplementary Text 2: Discussion of the concept of detailed balancing and feasibility of using EELS to measure temperature under temperature gradients” and Figure S6 have been added to address referee#3's concerns.
31. In Supplementary Information, “Supplementary Text 3: Discussion of the uncertainties in temperature measurement” have been added to address referee#2's concerns.
32. In Supplementary Information, the label in Figure S4c has been corrected as suggested by referee#2.

Response to the Referee's Comments

We would like to thank the referee for the highly constructive review concerning our manuscript. All the revisions to address referees' concerns are marked in red here. We believe that we have addressed all the concerns made by the referees (as detailed below), and made the revision accordingly.

Point to Point Responses

Reviewer #1

In my opinion, the authors have thoroughly addressed all the comments raised by the Referees during the review process. I appreciate the authors' comprehensive discussion in their response.

The manuscript has been significantly improved, and I believe it merits publication in Nature.

I have only one minor suggestion: the authors may consider restructuring the discussion of Figure 3 into multiple paragraphs for better readability. Specifically, I recommend:

1. Line 249: Begin a new paragraph with the sentence "For a three-phonon scattering process..."
2. Line 260: Begin a new paragraph with the sentence "On the cold side..."

Beyond this, I have no further objections.

The manuscript is excellent in every aspect—from Summary and Originality to Methods and Conclusions. I believe the scientific community will embrace this work as a breakthrough, demonstrating how electron microscopy can now also be used to characterize thermal transport at the nanoscale.

Response: We sincerely appreciate your positive assessment of our work. Following your recommendations, we have restructured the section organization to enhance the logical flow and clarity of the presentation.

Reviewer #2

I am pleased to see that the paper has satisfactorily addressed my comments. I have also read the authors' responses to other reviewers' comments and would like to share a few additional thoughts. The paper can be further enhanced by addressing the relatively minor points outlined below, after which I enthusiastically support its publication in Nature.

Response: We appreciate the reviewer's constructive comments. All suggestions have been carefully considered and incorporated into the revised manuscript.

1. In my view, this paper experimentally confirms phonon modal non-equilibrium and inelastic scattering, thanks to its unprecedented spatial resolution. The discussions of the underlying physical processes are generally sound. However, phonon interfacial scattering is a highly complex phenomenon, and additional transmission or scattering processes may be involved, so caution should be exercised in not prematurely ruling them out. For instance, it remains unclear from the existing literature whether the heat flux carried by interfacial modes is equivalent to the total inelastic heat flux across the interface. Furthermore, as the paper discusses, the interfacial region is typically on the order of 1-2 nm, which may be shorter than the wavelength of certain phonon modes, meaning these phonons may not "see" the interfacial region. For these reasons, it remains uncertain how many phonon modes scatter inelastically through the interfacial modes as a bridge, and how many scatter inelastically without involving the interfacial modes. The latter refers to phenomena where, in the case of three-phonon scattering, a phonon on one side directly scatters into two phonons on the other side, or, in the case of four-phonon scattering, two phonons on one side directly scatter into two new phonons on the other side. I very much like the authors' statement "This mechanism is applied to those phonon modes which require inelastic scattering through interface modes to transfer energy. However, for some modes with multiple-type scattering pathways, the scattering process and heat transfer mechanism should be much more complicated." My overall sense is that inelastic scattering processes involving interfacial phonons, as well as those not involving them, are likely co-existing and competing processes. Further studies will be needed to fully understand the role of interfacial modes. The authors have done an excellent job and can further strengthen the discussions by ensuring caution, particularly in addressing

more speculative aspects, which could be clarified.

Response 1: Thank you for your valuable suggestion. We fully agree with the reviewer's comment that "phonon interfacial scattering is a highly complex phenomenon." In the revised manuscript, we have refined the wording to adopt a more cautious and balanced tone in the sections that previously contained speculative statements. They read as follows (Page 9, Line 226):

"Based on these, we postulate a reasonable scattering mechanism underlying the non-equilibrium distribution of interface phonons."

Additionally, we have moved the discussion on the limitations of this transport model to the Discussion section (line 289), where we have provided a more detailed examination of these constraints. We have also discussed other possible scattering mechanisms including that you mentioned in your comment 4 below. The revised text is as follows (Page 11, Line 288):

"However, it should be noted that phonon transport in interface regions represents an inherently complex phenomenon, and whether the interface phonon-mediated heat flux equals to the total inelastic heat flux across the interface remains an open question. Alternative scattering pathways independent of interface modes, such as phonons with MFP exceeding the interface length scale or inelastic scattering processes only confined to bulk modes near the interface, may coexist or compete with the proposed mechanisms. Furthermore, higher-order phonon processes can also play a non-negligible role in thermal conductivity, particularly in those systems with significant mass-mismatch⁶. These aspects demand thorough investigation in future studies to fully characterize the underlying transport physics."

2. Previous theoretical development of modal non-equilibrium molecular dynamics (Refs. 20 and 22 in the current paper) predicted intrinsic phonon modal non-equilibrium and the existence of inelastic scattering across interfaces, but these predictions remained unvalidated for many years. One of the key contributions of the current paper is its experimental validation of these theories. To highlight this contribution of the current

paper and facilitate the storyline, the statement in the abstract "...yet the nanoscale dynamics remain largely unknown due to experimental limitations in measuring the temperature of the buried interface and resolving its non-equilibrium phonon distributions^{4–7}" can be modified to "...yet the nanoscale dynamics remain largely unknown due to experimental limitations in measuring the temperature of the buried interface and resolving its non-equilibrium phonon distributions^{4–7} predicted by theories^{20,22}", or something alike. Similarly, to acknowledge prior theoretical works and better position the current work, the statement in the introduction "Nevertheless, the phonon dynamics across buried interfaces during thermal transport still remain poorly understood, leaving several important issues unresolved such as the intrinsic interface temperature drop width, the temperature gradient induced non-equilibrium phonon distribution at the interface and evolution and reversibility of interface phonon during heat transfer" can be modified to "Nevertheless, the phonon dynamics across buried interfaces during thermal transport still remain poorly understood, leaving several important predictions unvalidated such as the intrinsic interface temperature drop width,^{^refs.xx} the temperature gradient induced non-equilibrium phonon distribution at the interface,^{^refs.20,22} and evolution and reversibility of interface phonon during heat transfer ^{^refs.zz}". I am quite sure that the experimental data presented in this paper will inspire further theoretical advancements, fostering a continuous loop of progress between theory and experiment.

Response 2: Thank you for the suggestions. We agree that this field has seen substantial prior theoretical work. We have adopted your suggestions in the revised abstract (Page 2, Line 28) and introduction (Page 4, Line 72).

"Although the interface phonon-mediated processes are theoretically established^{3–6} as the dominant mechanism for interfacial thermal transport in semiconductors⁷, their nanoscale dynamics remain experimentally elusive due to challenges in measuring the temperature and non-equilibrium phonon distributions across the buried interface^{8–11}."

"Nevertheless, the phonon dynamics across buried interfaces during thermal transport still remain poorly understood, leaving several important predictions unverified such as the intrinsic interface temperature drop width^{3,4}, the temperature gradient induced non-

equilibrium phonon distribution at the interface^{5,6} and evolution of interface phonons during heat transfer³⁰.”

3. In the abstract, the statement “...the interfacial inelastic scattering causes substantial non-equilibrium phonons nearby...” is confusing. In fact, scattering tends to bring the system to equilibrium instead of non-equilibrium. The mismatch of a phonon mode’s bulk thermal conductivity and interfacial conductance is a key cause of phonon modal non-equilibrium. Scattering then tends to reduce this non-equilibrium.

Response 3: Thank you for pointing out our oversight in the expression, and what you said is absolutely correct and exactly what we were trying to say before. In our proposed framework in the main text, scattering indeed serves as the primary mechanism facilitating the transition from non-equilibrium to equilibrium states. We have revised the description in Page 2, Line 34 (considering the length limit):

“During thermal transport, the mismatch of phonon modes’ thermal conductivity at the interface causes substantial non-equilibrium phonons nearby...”

4. Line 228-230: “...but only transfer energy through inelastic scattering with the interface modes...” How can we rule out the possibility that a phonon scatters inelastically on its own side, creating new phonons that transmit elastically across the interface? This possibility can be acknowledged if it cannot be ruled out.

Response 4: We appreciate your insightful observation regarding the possibility of “a phonon scatters inelastically on its own side, creating new phonons that transmit elastically across the interface”. We can reach a consensus that these modes must be transmitted across the interface through inelastic scattering. Indeed, such inelastic scattering may or may not involve interfacial modes. In the revised manuscript, we revised the description to be more inclusive in Page 8, Line 209:

“...most AlN bulk phonons in this energy interval become isolated modes, i.e., they cannot directly propagate through the interface (see Fig. S5), but only transfer energy through inelastic scattering including (but not limited to) interactions with the interface modes. The relative simplicity of their scattering pathways make these phonon modes

ideal for studying inelastic scattering mechanisms involving interface modes.”

Reviewer #3

I am very impressed with the thoroughness of the authors' responses. I do not see any need for further edits and feel the manuscript is suited for publication.

Response: We sincerely appreciate the reviewer's positive evaluation of our work. We are pleased that the contributions of this study have been recognized. In preparing the final version, we have carefully proofread the manuscript and ensured all data representations meet publication standards.

Response 3.3

Your comment concerning $\mathbf{e}_\lambda(k, \mathbf{q}) \cdot \mathbf{q}$ is mostly correct. Having the diffraction pattern displaced so that the minimum collection angle (or scattered momentum, \mathbf{q}_β) is larger than the convergence semi-angle (momentum uncertainty, \mathbf{q}_α) will still result in preferential sensitivity to eigenvectors along the selected scattering direction because the dot product is a projection. In other words, the convergence angle gives a momentum uncertainty perpendicular (and parallel) to the displacement direction such that if we consider the ratio of sensitivity to particular eigenvectors we get $s \approx \frac{\mathbf{e}_\lambda(k, \mathbf{q}) \cdot (\mathbf{q}_\beta \pm \mathbf{q}_\alpha)}{\mathbf{e}_\lambda(k, \mathbf{q}) \cdot (\pm \mathbf{q}_\alpha)} = \frac{\mathbf{e}_\lambda(k, \mathbf{q}) \cdot \mathbf{q}_\beta}{\mathbf{e}_\lambda(k, \mathbf{q}) \cdot (\pm \mathbf{q}_\alpha)} + 1$, so if $\mathbf{q}_\beta > \mathbf{q}_\alpha$ then we have more sensitivity to the modes parallel to \mathbf{q}_β than perpendicular to \mathbf{q}_β . You are correct that in the current framework of detailed balance occupation should not be impacted by the selection direction. The major point of the initial comment is that you may or may not see specific interface modes in a material depending on your off-axis selection as clearly shown in Hoglund *et al. Adv. Matt* (2024). Therefore you can only comment on the modes and scattering pathways visible within the current data, but cannot comment on the ensemble of all modes.